ecology, evolution

omnivory, macroevolution, mammalian evolution, diet evolution

**Author for correspondence:**
Dana M. Reuter
e-mail: dreuter@uoregon.edu

# What is a mammalian omnivore? Insights into terrestrial mammalian diet diversity, body mass and evolution

Dana M. Reuter[1,2], Samantha S. B. Hopkins[2,3,4] and Samantha A. Price[5]

[1]Florida Museum of Natural History, University of Florida, Gainesville, FL 32611, USA
[2]Department of Earth Sciences, and [3]Clark Honors College, 1272 University of Oregon, Eugene, OR 97403, USA
[4]Museum of Natural and Cultural History, 1680 East 15th Avenue, Eugene, Oregon 97403, USA
[5]Department of Biological Sciences, Clemson University, 132 Long Hall, Clemson, SC 29634, USA

DMR, 0000-0001-5637-7331; SSBH, 0000-0003-2289-7472; SAP, 0000-0002-1389-8521

Mammalian omnivores are a broad group of species that are often treated uniformly in ecological studies. Here, we incorporate omnivorous dietary differences to investigate previously found mammalian macroevolutionary and macroecological trends. We investigate the frequency with which vertebrate prey, invertebrate prey, fibrous plant material and non-fibrous plant material co-occur in the diets of terrestrial mammals. We quantify the body size distributions and phylogenetic signal of different omnivorous diets and use multistate reversible jump Markov chain Monte Carlo methods to assess the transition rates between diets on the mammalian phylogenetic tree. We find omnivores that consume all four food types are relatively rare, as most omnivores consume only invertebrate prey and non-fibrous plants. In addition, omnivores that only consume invertebrate prey, many of which are from Rodentia, are on average smaller than omnivores that incorporate vertebrate prey. Our transition models have high rates from invertivorous omnivory to herbivory, and from vertivory to prey mixing and ultimately invertivory. We suggest prey type is an important aspect of omnivore macro-evolution and macroecology, as it is correlated with body mass, evolutionary history and diet-related evolutionary transition rates. Future work should avoid lumping omnivores into one category given the ecological variety of omnivore diets and their strong evolutionary influence.

## 1. Introduction

We have long known that diet is intertwined with other aspects of mammalian evolution and ecology. Using three simple trophic categories: omnivory, carnivory and herbivory, a variety of macroecological and macroevolutionary patterns have been identified over the last few decades. Dietary type in mammals has been found to correlate with body size differences [1], life-history traits [2], tooth morphology [3], jaw morphology [4], digestive-tract morphology [5], diversification rates [6] and geographical distribution [7]. From these studies, we have learned that omnivores on average have intermediate tooth morphology [3] and intermediate body sizes [1] between herbivores and carnivores. We have also learned that mammalian omnivores have lower diversification rates than herbivores and carnivores [6]. Omnivory is often an 'evolutionary sink', with most of omnivore diversity coming from transitions into omnivory from other specialist dietary groups instead of within-guild speciation [6]. This pattern has been found not only in mammals but in birds as well [8]. While our understanding of how diet influences evolution and ecology has improved, we still have limited knowledge of what constitutes a mammalian omnivore and how differences in omnivore ecology influence these macroevolutionary findings.

Omnivores are usually defined as animals that consume both plants and animals. They are often considered generalists in terms of being able to gain

substantial energy and nutrition from a variety of plant and animal sources; however, they can vary in their degree of dietary specialization and food mixing. It has been observed that most mammals are not complete generalists and only combine certain food materials, such as fruit and animal material or fruit and foliage, because it would be physiologically difficult to digest all food types [5]. Many taxon-specific studies have also shown omnivores specialize in eating specific food items, sometimes at particular times of the year (e.g. [9–13]). However, differences in specialization and food mixing among omnivores have been understudied in macroevolutionary studies, which leave open questions for evolutionary biologists. We know that there is lineage and diet type variation in macroevolutionary trends among the three basic trophic groups [8]. For instance, with respect to body mass trends in carnivorous mammals, studies have found that insectivorous mammals are smaller than vertivorous predators [14,15]. In addition, finer dietary categorization has revealed that mixed-feeding ruminants had higher diversification rates than browsing ruminants [16]. Despite these successes in unpacking other diet categories, omnivory has mostly been left untouched even though dietary variation has been well documented for many mammalian omnivores [9–13]. Important information from ecological and phylogenetic comparative studies can be gained when omnivory is redefined to encompass more detailed dietary categorizations [17]. In this study, we further investigate the evolution of mammalian omnivory by quantifying: (i) which food materials are most often eaten together among mammals, (ii) how omnivorous dietary strategies are distributed on the mammalian phylogeny, (iii) the correlation between omnivorous diet type and patterns in mammalian body mass, and (iv) the transition rates into and out of mammalian omnivore dietary states. These objectives are crucial for building our basic knowledge of omnivore macroecology and macroevolution. Understanding the covariation in foods mammalian omnivores consume will expand our understanding of the macroevolutionary limitations of mixing food materials. A deeper look at the relationship between body size and type of omnivory will let us test if the body mass patterns we see in specialist groups occur in food-mixing lineages. Knowing how omnivorous strategies are distributed across the mammalian phylogeny will help us to understand how omnivory evolves through time across different lineages. Finally, including more detailed diet information when estimating the dietary transitions that have occurred over the evolutionary history of mammals will allow us to identify which diets are acting as long-term strategies, temporary states or evolutionary sinks.

## 2. Methods

### (a) Dataset and phylogenetic tree

Using previously published datasets, we compiled diet data and body masses for 1437 extant terrestrial mammals (about a quarter of all mammal species) (electronic supplementary material, data S1). We chose to exclude aquatic mammals, as they are dependent on food webs with a dramatically different structure from their terrestrial counterparts and are expected to experience different ecological and evolutionary pressures compared with terrestrial mammals [18]. For this study, we chose to use a dataset compiled from primary literature sources generated previously by two of the authors of the present paper [6]. The results of any diet study are determined in part by the choice of diet data and the way those data are standardized. We used the

**Table 1.** Number of species found in each diet category used in this study. Four food categories were used to determine diet type: invertebrate prey, vertebrate prey, non-fibrous plant parts and fibrous plant parts.

| diet guilds | trophic level | no. species |
|---|---|---|
| non-fibrous/fibrous | herbivore | 316 |
| invertivore | carnivore | 263 |
| fibrous | herbivore | 160 |
| non-fibrous | herbivore | 158 |
| invertivorous/non-fibrous/fibrous | omnivore | 144 |
| invertivorous/non-fibrous | omnivore | 136 |
| vertivore/invertivore | carnivore | 86 |
| vertivorous/invertivorous/non-fibrous | omnivore | 69 |
| vertivorous/invertivorous/non-fibrous/fibrous | omnivore | 41 |
| vertivore | carnivore | 36 |
| invertivorous/fibrous | omnivore | 8 |
| vertivorous/invertivorous/fibrous | omnivore | 7 |
| vertivorous/non-fibrous | omnivore | 7 |
| vertivorous/non-fibrous/fibrous | omnivore | 5 |
| vertivorous/fibrous | omnivore | 1 |

Price *et al.* [6] data because they are based on observational data at the species level, and provide a basis for comparison with prior analyses of the effect of diet on diversification histories [6]. This dataset does not contain dietary inferences based on phylogeny or morphology, and was compiled using studies that reported stomach contents or cheek-pouch contents, the contents of food stores, direct behavioural observations or faecal analysis. This dataset codes the presence of a particular food in the diet of an animal if the diet description indicated the food was regularly consumed by individuals of that species, or it constituted at least 5% of the food consumed by volume, weight or feeding time (see Price *et al.* [6] for more details of coding). The dataset has broad taxonomic and phylogenetic representation and good coverage of all the mammalian orders (electronic supplementary material, data S1). As it was compiled, species from under-represented families in the diet literature were prioritized, as opposed to obtaining exhaustive samples of well-represented genera [6]. Because of the high computational demands of using multiple categories, the data in Price *et al.* [6] were originally analysed using the three basic trophic categories. They were, however, originally collected using four food categories: invertebrate prey, vertebrate prey, fibrous plant parts (mature leaves, stems, wood and bark) and non-fibrous plant parts (any other parts of plants). We used these four food types to assign each species to 1 of 15 diet guilds (table 1). Body masses for omnivorous species were gathered from the PanTHERIA database [19]. For all phylogenetically informed analyses, we used a fully resolved set of phylogenetic trees from Faurby & Svenning [20].

### (b) Omnivore body mass

To understand the relationship between body size and diet in omnivorous mammals ($n = 418$), we performed an ANOVA comparing the natural-logged body masses and a Tukey–Kramer test for unequal sample sizes (stats package), in the statistical program R [21]. We also checked for equality of variance between

**Table 2.** BayesTraits categories and phylogenetic signal results. The $D$ statistic is close to 1 if the observed trait has a phylogenetically random distribution, or 0 if the observed trait is dispersed on the tree in a way that is consistent with a threshold model of Brownian motion evolution. Values lower than 0 indicate phylogenetic clustering.

| simplified diet guilds used as BayesTraits categories | no. species | phylogenetic signal — mean $D$ ± mean s.d. |
|---|---|---|
| herbivore | 634 | $0.030 \pm 0.005$[a] |
| invertivorous omnivore | 288 | $0.461 \pm 0.003$[b] |
| invertivore | 263 | $-0.072 \pm 0.007$[a] |
| vertivorous/ invertivorous omnivore | 117 | $0.505 \pm 0.004$[b] |
| vertivore/invertivore | 86 | $0.440 \pm 0.004$[b] |
| vertivore | 36 | $0.096 \pm 0.014$[a] |
| vertivorous omnivore | 13 | $0.813 \pm 0.026$[b] |

[a]Statistically different from a random distribution of states.
[b]Statistically different from both a random distribution and a distribution consistent with a Brownian motion model of evolution. More detailed results can be found in electronic supplementary material, data S2.

groups using the *leveneTest* function from the car package in R [22]. To account for phylogenetic autocorrelation, we also performed PGLS ANOVAs implemented by using functions in the caper package [23].

## (c) Phylogenetic signal

We calculated the phylogenetic signal of each diet category, treating each diet category as a binary trait [24] over 10 randomly selected trees with the *phylo.d* function in the caper package in R [23]. This method calculates a $D$ statistic that is close to 1 if the observed trait has a phylogenetically random distribution or 0 if the observed trait is dispersed on the tree in a way that is consistent with a threshold model of Brownian motion evolution [24]. The trait distribution for the Brownian motion model is calculated by simulating a continuous trait along the phylogeny, defining a threshold value that ensures that the number of tips with each character state remains the same as in the original dataset, then defining the character state at each tip using the threshold value of the continuous trait. Values lower than 0 indicate phylogenetic clustering beyond what is expected by the Brownian motion threshold model. The *phylo.d* function also tests for significant departure from both a phylogenetically random distribution and the phylogenetic distribution generated under the threshold model.

## (d) Transition rates

We simplified the dietary guilds to represent only differences in prey type as the body mass distributions in omnivores grouped by plant material consumed were not significantly different (see table 2 for the seven simplified categories by prey type and number of species). We used these simplified categories in our other analyses to greatly increase our statistical power and decrease our computational time. In addition, we chose to use the body mass patterns as a determining factor for merging dietary guilds because body mass is directly linked to energetic

needs and is an important ecological factor that many other traits covary with [25]. We acknowledge that body mass is just one trait that could differ between omnivorous mammals, and other traits, such as tooth morphology or jaw morphology, could be used to justify other groupings. This simplification also resulted in there being one guild for herbivores instead of three. We used one guild for herbivores to be consistent with the omnivore guilds and to allow comparison with the results of Price *et al.* [6].

We calculated transition rates between the seven dietary guilds using Bayesian Markov chain Monte Carlo (MCMC) methods in the program BayesTraits [26]. Specifically, a multi-state reversible jump MCMC was used to estimate transition rates without assuming a single model of trait evolution [27]. Reversible jump MCMC explores all possible models and generates a posterior distribution of models and parameter estimates by setting each transition parameter either to a unique value, or equal to another transition rate, or to 0. This process allows the exploration of log-likelihood especially when there is a large number of possible models. Because this is computationally intensive, all BayesTraits analyses were run on the University of Oregon Talapas High Performance Computing cluster.

To take into account variability in tree topology, we ran independent chains on 100 randomly selected fully resolved trees [28]. We used uniform hyperpriors to seed the exponential prior on the parameters, seeding the mean of the exponential prior from a uniform distribution over the interval 0–10 or 0–2. To ensure stationarity was reached each chain was run for 1 billion iterations with a sampling interval of 300 000 and a burn-in of 100 000 iterations. We examined the effective sample sizes, autocorrelation and convergence using packages coda and btw in R (electronic supplementary material, data S3) [29,30]. We also checked the autotuning mechanism by examining schedule files to make sure the chains were mixing appropriately. The medians and interquartile ranges (IQRs) were then calculated for each transition rate along with the frequency with which a transition rate was reconstructed as zero (% Z). To investigate whether the differences in transition rates were meaningful, we ran the same analyses on a tree with randomly reassigned dietary categories. We produced a random dataset in R using the *sample* function on our existing data to guarantee the same number of individuals in each dietary guild. We then used the same reversible jump MCMC procedure in BayesTraits to calculate median transition rates, % Z, and model posterior distribution. This allowed us to determine whether our observed results differed from those expected when there is no phylogenetic signal in dietary guilds.

Additionally, to accurately compare our transition rates with those estimated by Price *et al.* [6], which used a different phylogeny and methods, we used the same reversible jump MCMC procedure in BayesTraits to estimate transition rates between the three main trophic categories (herbivory, carnivory and omnivory). We ran 10 independent chains using 10 randomly selected trees. Furthermore, following the procedures of Price *et al.* [6], we conducted analyses using the 'multiple state speciation extinction' (MuSSE) model in the diversitree package [31,32] on our seven dietary guilds. We estimated model fit using maximum likelihood instead of Bayesian MCMC methods because of computational constraints. We ran two models, one with the speciation and extinction rates constrained to those reported in Price *et al.* [6] and one with no constraints. We incorporated the sampling frequency to be 0.22 of all Mammalia as the current reported number of mammalian species is 6495 [33]. We ran each model on 10 randomly selected trees.

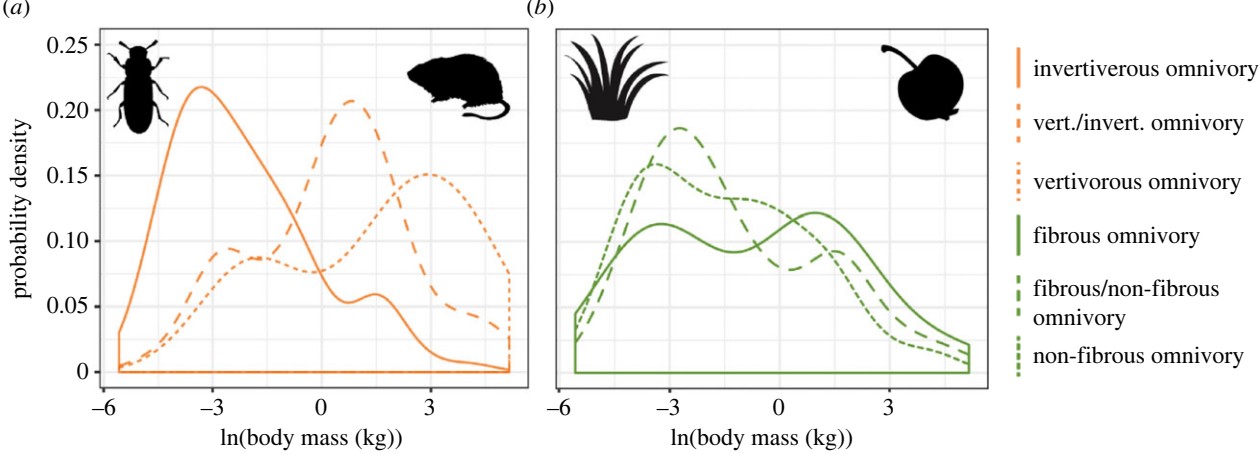

**Figure 1.** Omnivore body mass distributions are separated by diet type.

**Table 3.** Omnivore body mass distributions.

| omnivore diet category | body mass (kg), mean (range) | ln(body mass (kg)), mean (range) |
|---|---|---|
| *grouped by plant material* | | |
| fibrous omnivore | 8.65 (0.012 to 108.4) | −0.72 (−4.40 to 4.69) |
| fibrous/non-fibrous omnivore | 5.82 (0.004 to 172.7) | −1.22 (−5.43 to 5.15) |
| non-fibrous omnivore | 3.21 (0.003 to 99.9) | −1.47 (−5.57 to 4.60) |
| *grouped by prey type* | | |
| invertivorous omnivore | 1.51 (0.004 to 93.1) | −2.10 (−5.57 to 4.53) |
| vertivorous omnivore | 23.09 (0.073 to 108.4) | 1.45 (−2.62 to 4.69) |
| vertivorous/invertivorous omnivore | 10.17 (0.007 to 172.7) | 0.28 (−5.01 to 5.15) |

## 3. Results

### (a) Dietary guild diversity

There are large differences in species richness among the mammalian diet guilds (table 1). Herbivores that eat both non-fibrous and fibrous plant material are the most diverse ($n = 316$, 22% of sample) followed by invertivores ($n = 263$, 18% of sample). The omnivorous diet strategy with the highest species richness is the one that combines these two diets, consuming a mix of invertebrate prey and both non-fibrous and fibrous plant material ($n = 144$, 10% of sample), but it is closely followed by those consuming a mix of invertebrate prey and only non-fibrous plant material ($n = 136$, 9% of sample). Predators that eat both vertebrate and invertebrate prey are more diverse ($n = 86$, 6% of sample) than any omnivorous strategy that incorporates vertebrate prey. Mixing all four food types ($n = 41$, 3% of sample) only has slightly higher species richness than vertivory ($n = 36$, 3% of sample). Five dietary categories contain fewer than 10 species, suggesting these diets are rare in Mammalia. These categories mix fibrous plant material with either invertebrate or vertebrate prey or vertebrate prey with non-fibrous plant material. The least occupied dietary guild is the vertivorous/fibrous omnivore, which eats only vertebrate prey and fibrous plants; the panda, *Ailuropoda melanoleuca*, is the sole member. The panda eats mostly fibrous plants (bamboo leaves and shoots) but also consumes vertebrate prey in the form of rodents and other small vertebrates [34]; however, it seems to be alone in its dietary habits.

When our dataset is sorted into groups separated by animal prey type (table 2), diversity pattern differences among omnivorous strategies still exist. Invertebrate omnivory is the second most diverse diet type ($n = 288$, 20% of sample) on the mammalian tree after herbivory ($n = 634$, 44% of sample). The diet guild with the lowest species richness mixes vertebrate prey with plant material ($n = 13$, 1% of sample). Examples of species with these unique diets are: *Chrysocyon brachyurus*, *A. melanoleuca* and *Ailurus fulgens* [9,34,35].

### (b) Omnivore body mass

When we compared body mass distributions among the different omnivore guilds, we found that lower size ranges are similar across all groups, but the omnivores that eat a mix of all food materials have a larger upper body mass limit. The largest omnivore is *Ursus arctos* (vertivorous/invertivorous omnivore 172 kg) and the smallest is *Sorex trowbridgii* (invertivorous omnivore 3.8 g). Although the diet groups have similar body mass ranges, when omnivores are grouped by prey type, they have very different distributions (figure 1). For instance, although there are a few large omnivores that specialize on insects, such as the sloth bear *Melursus ursinus* [12], most invertivorous omnivores are small (mean = 1.51 kg, table 3). In fact, most invertivorous omnivores are much smaller than omnivores that consume vertebrate prey, with vertivorous/invertivorous omnivores having intermediate body masses (mean = 10.17 kg) and vertivorous omnivores having the largest mean body mass (mean = 23.09 kg). Table 3 and figure 1

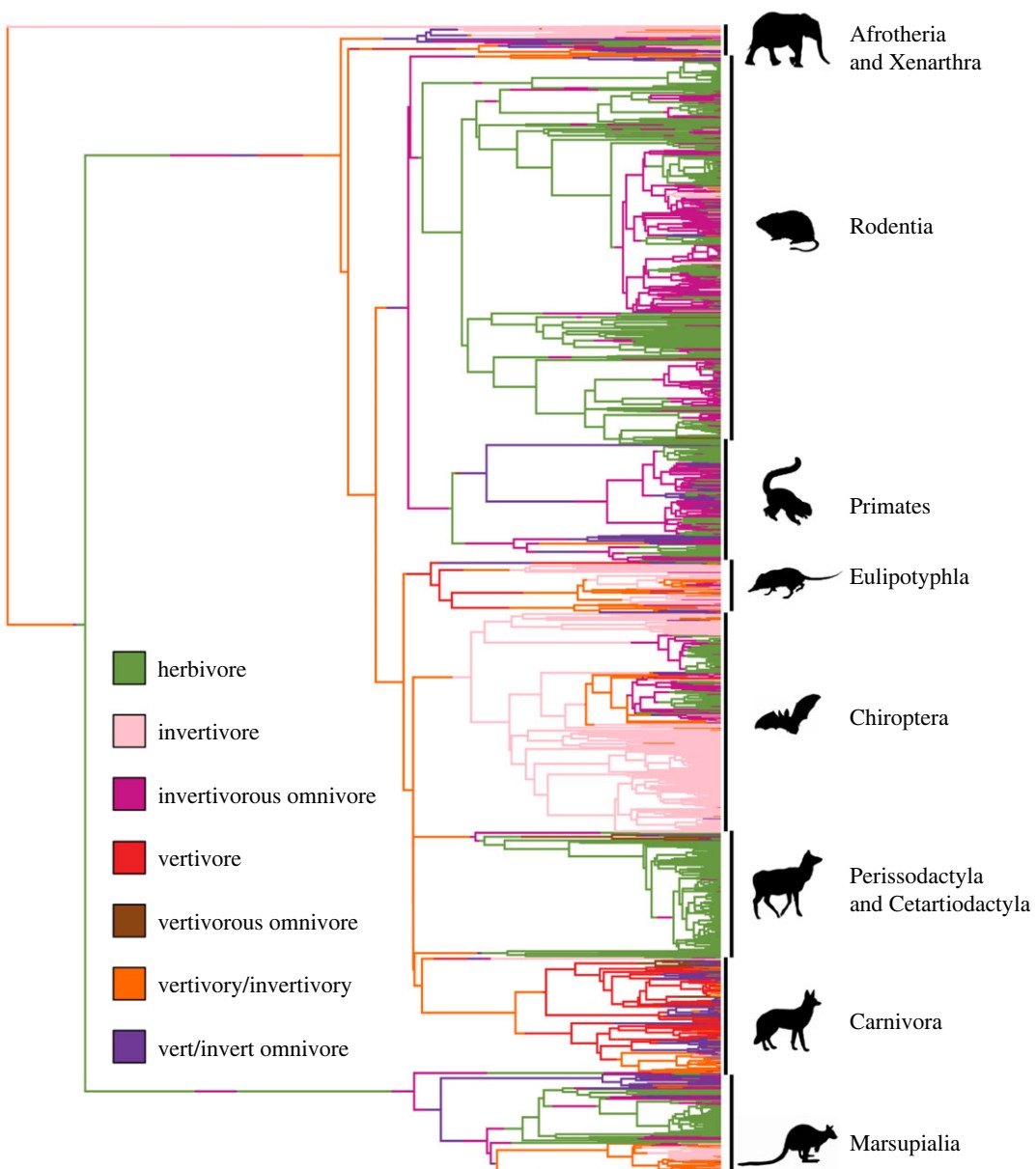

**Figure 2.** Diet distributions on the mammalian phylogeny. Branch colour corresponds to diet. Ancestral diet states on the branches were reconstructed using the *make.simmap* function from the phytools package. Ancestral states were reconstructed to highlight the distribution and diversity of mammalian diets among the 1437 species of mammals used for this study. Some diet types, such as invertivory and herbivory, are highly clustered, while omnivory is more distributed along the tree. The branch state reconstruction was done for the purposes of illustration and is not an accurate representation of all the transitions that were calculated using BayesTraits or diversitree.

also show that when omnivores are grouped by type of plant material consumed the groups have similar body mass ranges and distributions to each other.

We found no significant difference in variances between groups via Levene's tests (grouped by prey, $F = 1.15$, $p = 0.32$; grouped by plant material, $F = 1.14$, $p = 0.32$). Our ANOVA results confirm that when omnivores are grouped by prey type, there are significant differences between their mean body masses ($F = 66.89$, $p < 0.00001$) but there is not a significant difference when omnivores are grouped by plant material consumed ($F = 1.08$, $p = 0.34$). Pairwise comparisons reveal omnivores that consume only invertebrate prey have a significantly lower average body mass than both groups of omnivores that consume vertebrate prey ($p < 0.00001$ for both comparisons). There was not a significant difference between omnivores that eat both prey types and omnivores that only eat vertebrate prey ($p = 0.13$). The pairwise tests combined with the body mass distributions in figure 1 suggest that

most invertivorous omnivores are much smaller than omnivores that include vertebrate prey in their diets despite the body mass ranges being similar. Our PGLS ANOVA, however, indicates that the differences found between omnivores that consume only invertebrate prey and omnivores that consume vertebrate prey are most likely driven by phylogeny. We found no significant difference between the body masses of omnivores when omnivores are grouped by prey type ($F = 0.797$, $p = 0.45$). A considerable number of omnivores that consume only invertebrate prey are found within Rodentia and many vertivorous/invertivorous omnivores are found within Carnivora (figures 1 and 2).

## (c) Phylogenetic signal

We found that herbivores, invertivores and vertivores have phylogenetic signal consistent with a Brownian motion threshold model of evolution (table 2). We also found that

omnivores and mixed-feeding dietary guilds, such as mammals that consume both invertebrate and vertebrate prey, have a phylogenetic distribution that is more dispersed than the Brownian motion threshold model but is clustered more than expected under the random model. These intermediate phylogenetic signal values were also found for mixed feeders (e.g. mammals that consume both fibrous and non-fibrous plant material) and omnivores when guilds were defined by plant material consumed (electronic supplementary material, data S2). These phylogenetic signal values suggest that mixed feeders have multiple origins on the mammalian tree and are not as phylogenetically conserved as herbivores, invertivores and vertivores (figure 2). Omnivores that only eat vertebrate prey had the highest $D$ estimate ($D = 0.813 \pm 0.026$, mean ± s.d.), showing that they are the most dispersed on the tree, while invertivores were the most phylogenetically clustered, with the lowest estimate of $D$ ($D = -0.072 \pm 0.007$, mean ± s.d.). It is worth noting that $D$ is most powerful with sample sizes of 50 and above [24], and our vertivores and omnivores that only eat vertebrate prey have sample sizes below this, although standard deviation values for both were low.

### (d) Transition rates

In our BayesTraits reversible jump MCMC results, we confirmed that placing different constraints on the uniform hyperprior interval made little difference to the transition rate estimates, as both hyperprior intervals (0–10 and 0–2) converged on similar average rates (electronic supplementary material, data S4 and figures S1 and S2). Our randomized dataset produced overall higher median transition rates and higher IQRs (electronic supplementary material, data S5 and figure S3) than the empirical data, which is consistent with the phylogenetic signal within our dataset having a substantial impact. Our BayesTraits transition rate estimates between the three main trophic categories (herbivory, carnivory and omnivory) show that transitions between herbivory and carnivory are almost zero (electronic supplementary material, table S1). The highest transition rate was from omnivory to herbivory ($1.4712 \pm 0.6098$, median ± IQR). The other transitions into and out of omnivory had lower transition rate estimates (figure 3).

Our BayesTraits reversible jump MCMC analyses using the seven simplified diet categories show low to non-existent transition rates between herbivory, invertivory and vertivory (table 4). There are low transition rates out of herbivory and invertivory and many of these rates are estimated as zero in 90% of the models (electronic supplementary material, table S2). Our results also indicate transitions to a new food type have intermediate steps through omnivory or mixed feeding (figure 3). We also found that the invertebrate omnivore guild has high transition rates into herbivory, while other guilds do not (figure 3). Transition rates out of vertivory were high for transitions into vertivory/invertivory and vertivorous/invertivorous omnivory. Some high median transition rates between omnivorous and mixed-feeding guilds have high IQRs (and hence are poorly constrained), such as the transition between vertivory/invertivory and vertivorous/invertivorous omnivory. There are also quite a few intermediate transition rates that are well constrained, such as the transition from invertivorous omnivory to invertivory.

Our MuSSE results (electronic supplementary material, data S6; figure 3) generally agreed with our BayesTraits reversible jump MCMC estimates. When speciation and extinction were unconstrained many of the estimated transitions were consistent with our BayesTraits results, although many were poorly estimated and had high IQRs (electronic supplementary material, figures S4–S6). A few key differences between the models do stand out. When speciation and extinction are constrained to those estimated by Price et al. [6], the highest transition rate is from vertivory to vertivorous/invertivorous omnivory. Additionally, in the constrained MuSSE, the median estimate for the transition rate from herbivory to invertivorous omnivory is higher than the transition rate from invertivorous omnivory to herbivory. This high transition rate from herbivory is not present in the reversible jump MCMC results or the unconstrained MuSSE results.

## 4. Discussion

Our findings reveal that, although macroevolutionary differences exist among the three trophic groups (herbivory, omnivory and carnivory), there are macroecological and macroevolutionary patterns within omnivory that have been previously overlooked. Within mammalian omnivory, diet type is reflected in patterns of diversity, body mass, phylogeny and evolutionary transition rates. Our results suggest that prey type plays an important role in understanding omnivore macroevolution and ecology, just as it does across mammalian trophic groups.

In terms of diversity, in our dataset, the two most species-rich mammalian omnivore guilds only consume invertebrates as their animal food source, suggesting that most omnivorous mammals rely on invertebrates and not vertebrate prey. In addition, combining animal material with fibrous plant material is not a very common dietary strategy, agreeing with past observations that eating fibrous material with animal protein is physiologically difficult and evolutionarily rare [5]. In fact, even mixing all four food types (non-fibrous plants, fibrous plants, invertebrates and vertebrates) is a relatively uncommon strategy, characterizing only around 10% of omnivores in our dataset. These mammals include some ursids, canids, primates, rodents and suids. This finding is consistent with previous work, which highlighted the rarity of species that eat equal parts green plants, fruit, seeds, invertebrates and vertebrates within mammals [17]. When we think of omnivorous mammals, highly generalist species that mix all food types often come to mind; however, our results reveal that this strategy is rare. Our findings support previous work which concluded that omnivorous diets have several distinct diet types [17]. Most omnivores are consuming non-fibrous plant material, making invertebrates and non-fibrous plant material the most common food mixing in Mammalia. Many of these mammals are from within Rodentia, Primates and Chiroptera. Insects and non-fibrous plant material are easily found and consumed together, as many insects consume and inhabit fruits and seeds. Past work found that insect-infested fruit was a preferred food source by many primates, possibly because it is a protein- and carbohydrate-rich food source [36]. It seems that past suggestions that most omnivores consume fruits and seeds because of ease of access and digestibility hold true [5,37]. Our work also uncovers the equal importance of invertebrates in

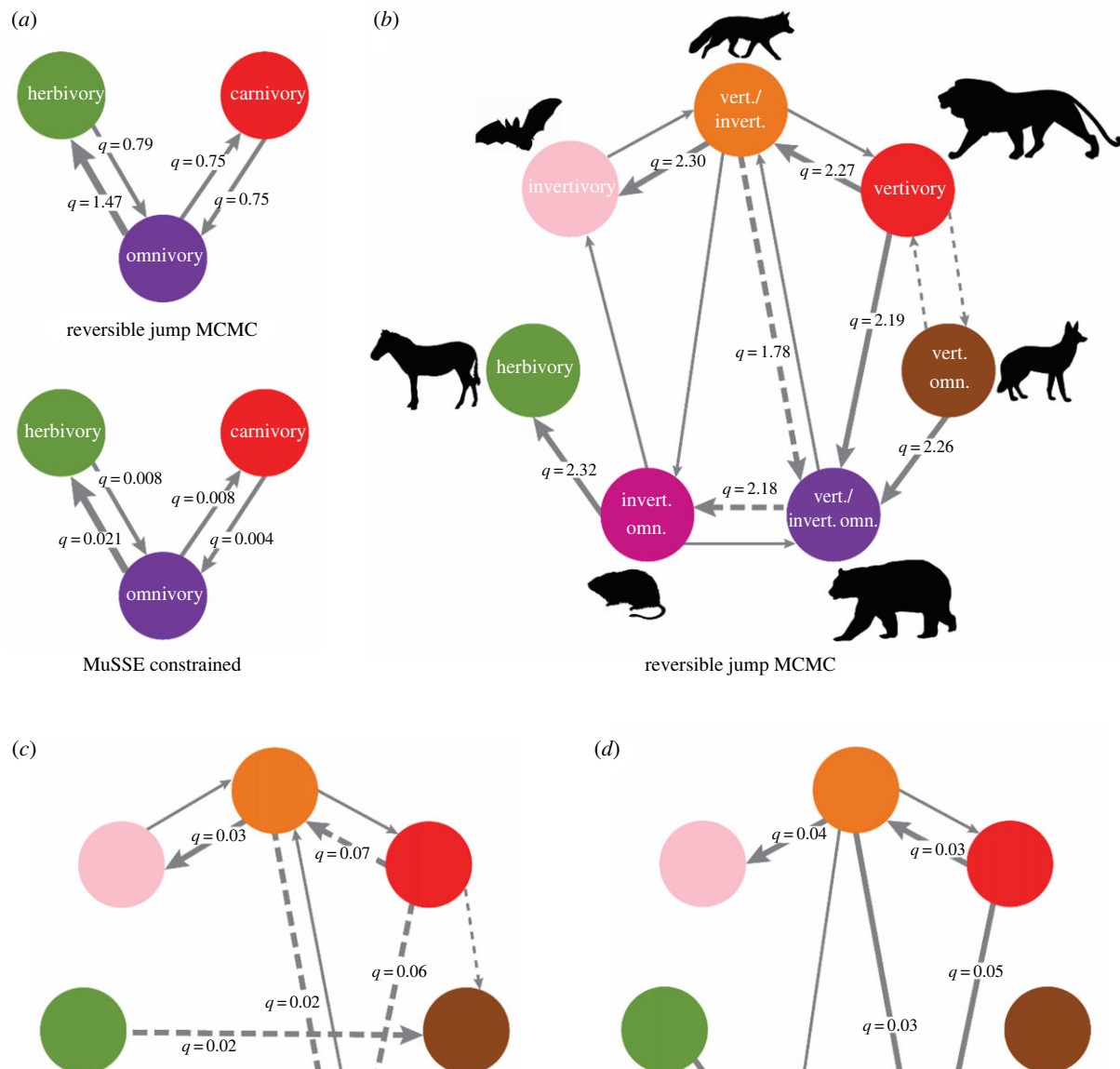

**Figure 3.** A summary of transition rates between dietary groups. Parameter $q$ represents the median transition rate estimated from the posterior distribution of models. Thick arrows represent high transition rates. Smaller arrows represent lower transition rates. Dashed arrows indicate that the IQR for that transition rate is high. (a) Median transition rates that were estimated when just using the three basic trophic levels. The reversible jump MCMC results are from this study and the MuSSE constrained are the results from Price *et al.* [6] where extinction and speciation were both constrained. (b) Summary of the transition rates estimated using reversible jump MCMC results among seven dietary categories. (c) MuSSE model where speciation and extinction where unconstrained. It should be noted that the unconstrained MuSSE model had higher IQRs for many of the higher transition rates (see electronic supplementary material, data S6). (d) MuSSE model where speciation and extinction where constrained to the values of the best-fit model of Price *et al.* [6].

omnivorous diets, which should not be overlooked when studying mammalian omnivores.

Our results also reveal that there is a relationship between omnivore prey selection and body mass. We found that omnivores specializing on invertebrate prey are on average smaller (mean body mass of 1.51 kg) than omnivores that incorporate vertebrate prey into their diet (mean body mass of 10.17 kg for vertivorous/invertivorous omnivory and 23.09 kg for vertivorous omnivory). Carbone *et al.* [14] estimated that the maximum sustainable mass for insectivores is around 21.5 kg and that the transition from small to large prey occurs around this mass as well. For our dataset, 21.5 kg is

around the mean body mass for omnivores that only incorporate vertebrate prey, and most omnivores that incorporate invertebrate prey are below this mass. Omnivores should be less energetically constrained by their prey because they are also relying on plant food sources for their energetic needs. However, our findings highlight that the overall trend previously found in the order Carnivora [14] and non-volant terrestrial mammals [15] is still detectable when examining the body masses of mammalian omnivores. It is evident that most invertivorous omnivores are smaller than omnivores that incorporate vertebrate prey, and most are below the maximum sustainable body mass of 21.5 kg for specialist

**Table 4.** Median transition rates ± IQR from BayesTraits reversible jump MCMC chains with an exponential prior 0, 2. Transitions are happening from first column to column headers. The values in italic type are higher transition rates and are represented in figure 3b. n.a., not applicable.

| | herbivore | invert. | invert. omn. | vertivore | vert. omn. | vert./invert. | vert./invert. omn. |
|---|---|---|---|---|---|---|---|
| herbivore | n.a. | 0 ± 0 | *0.7166 ± 0.1338* | 0 ± 0 | 0.0790 ± 0.0571 | 0 ± 0 | 0 ± 0 |
| invertivore | 0 ± 0 | n.a. | 0.0793 ± 0.0623 | 0 ± 0 | 0 ± 0 | *0.6945 ± 0.1361* | 0 ± 0 |
| invert. omn. | *2.3193 ± 0.3704* | *0.6792 ± 0.1534* | n.a. | 0 ± 0 | 0.0420 ± 0.0720 | 0 ± 0.0484 | *0.7168 ± 0.1340* |
| vertivore | 0 ± 0.0595 | 0 ± 0.0775 | 0 ± 0.0790 | n.a. | 0.5506 ± 0.6444 | *2.2681 ± 0.4289* | *2.1878 ± 0.5934* |
| vert. omn. | 0.0885 ± 0.7246 | 0.0499 ± 0.1468 | 0.0878 ± 0.7775 | *0.6409 ± 0.8278* | n.a. | 0.0664 ± 0.6094 | *2.2586 ± 0.4514* |
| vert./invert. | 0 ± 0.0430 | *2.3009 ± 0.3813* | *0.6961 ± 0.1827* | *0.7409 ± 0.1929* | 0 ± 0.0570 | n.a. | *1.7794 ± 1.5314* |
| vert./invert. omn. | 0.0867 ± 0.1206 | 0.0411 ± 0.0906 | *2.1752 ± 1.3540* | 0.1326 ± 0.5881 | 0.0550 ± 0.1500 | *0.7420 ± 0.1909* | n.a. |

insectivores. Additionally, our Tukey–Kramer pairwise comparisons indicate that omnivores that only eat invertebrate prey are smaller than both mixed-prey-feeding omnivores and omnivores that only eat vertebrate prey. This result further suggests that incorporating vertebrate prey as an omnivore requires a larger body mass just as it does for purely carnivorous mammals. However, our PGLS ANOVA results reveal that the relationship between body mass and diet in omnivores is influenced by evolutionary history. For instance, over half of the invertivorous omnivores are found within Rodentia ($n = 161$ out of 288). Our PGLS result, combined with our finding that omnivorous diets are over-dispersed on the mammalian phylogeny (relative to Brownian motion expectations), suggests that the effects of diet and phylogeny are intertwined across mammals and influence all other related traits (i.e. body mass). Our results show that the interconnected nature of diet, phylogeny and body mass is present even for omnivorous mammals, which have many unique origins on the phylogeny.

By contrast with the prey type correlations, we did not find a difference in body mass between incorporating fibrous plant material versus non-fibrous plant material. This result suggests that the type of plant material consumed does not constrain body mass in the same way that prey selection does. Although we did find some variation in phylogenetic signal (electronic supplementary material, data S2) related to plant material consumed, these differences were neither as large nor as significant as the differences related to prey type. Our results are not surprising given that the nature of the relationship between body mass and fibrous plant material utilization is unclear. Originally, fibre content was thought to scale with body mass because of decreasing digestibility [38]. Other studies have highlighted that small mammals have the capacity to digest fibrous material [39] and that there are inconsistencies with the proposed body mass pattern [40]. Our dataset contains many small mammals that combine fibrous plant material with other food sources, and our results confirm that omnivores of various sizes and phylogenetic history consume fibrous and non-fibrous plant material. We suggest that omnivore body mass does not reflect the earlier proposed energetic and physiological constraints of consuming fibrous material [38] and instead is more consistent with research suggesting that it might be a question of access and availability rather than digestibility [39,40]. It seems that omnivores are released from some of the proposed body size constraints of consuming fibrous material, possibly because plant material is not always the main food source. Our work highlights the importance of considering prey type when investigating mammalian diets. By contrast, past work has noted the importance of differentiating plant material for understanding mammalian ecology and evolution. These works have shown that frugivores and granivores differ in body mass [15], tooth morphology can differ based on amount of plant material consumed [3] and diversification rates in ungulates are related to type of plant material consumed [16]. Therefore, a more detailed dataset of plant consumption that includes relative amounts as well as material properties might reveal more subtle relationships between diet composition and body size within mammalian omnivores, similar to the relationships we uncovered with respect to prey selection.

Our transition rate analyses further highlight that lumping omnivores into one trophic guild can mask macroevolutionary trends. When omnivory is broken down into more ecologically and evolutionarily meaningful categories, mammalian evolution is shown to follow distinct trends, with prey type being an important determining factor. When estimating transition rates between the three basic trophic guilds (herbivory, carnivory and omnivory), our reversible jump MCMC results were generally similar to the Price *et al*. [6] rates. However, the highest transition rate was reversed, our new analyses found the highest rate of transition was from omnivory to herbivory while the best-fit model from Price *et al*. [6] estimated that the transition rate from herbivory to omnivory was highest. This is most likely caused by differences in the models used. BayesTraits does not take into account speciation or extinction, while the MuSSE model used by Price *et al*. [6] does. In fact, our transition estimates between the three basic trophic guilds are most similar to their model where both speciation and extinction rate are constrained, which resulted in the highest transition rate being from omnivory to herbivory. Our reversible jump MCMC models and MuSSE models that estimated transition rates between the seven detailed dietary guilds show that the three trophic guilds do not have enough detail to capture the nuanced transitions among the diet types, possibly explaining some of this difference. The high transition rate from herbivory observed in previous analyses

[6] is not present in the reversible jump MCMC results or the unconstrained MuSSE results and only is present when speciation and extinction are constrained to the values found by Price *et al*. [6]. Additionally, in our models, the only transitions out of herbivory happen into invertivorous omnivory. These transitions mainly happen in Rodentia and Primates. This result shows that while herbivorous lineages may lead to many omnivorous species, as was suggested by Price *et al*. [6], they are constrained by prey type and potentially small bodied. Our models also show that transition rates are lower out of invertivory, and higher transition rates exist out of vertivory and into omnivory and prey mixing. The models highlight two main evolutionary pathways, one from vertivory into increasingly invertivorous omnivory and ultimately herbivory, and one from vertivory to prey mixing and ultimately invertivory.

Additionally, we found that herbivory and invertivory are phylogenetically clustered. The combined patterns of high diversity, high phylogenetic clustering and low transition rates into other diet types suggest that herbivory and invertivory are highly successful and evolutionarily stable strategies, which may facilitate low extinction rates and/or high speciation rates. Vertivores, despite also being clustered on the mammalian phylogeny, have higher transition rates into mixed-feeding strategies and omnivory. Specifically, our results show that the transitions into omnivory and prey mixing occur at higher rates from vertivores and lower rates from herbivores and invertivores. The phylogenetic signal found in omnivores suggests that omnivorous strategies are dispersed over the phylogeny and their diversity is the result of these high rates of transition from vertivory and low rates of transition from herbivory, as opposed to diversification within omnivorous lineages. This insight agrees with past work on both mammals [6] and birds [8] which showed that omnivory has low rates of diversification and can act as an evolutionary sink. Our results demonstrate that important differences exist between omnivorous groups and their evolutionary origins when prey type is incorporated into evolutionary models.

Our results also reveal that transitions out of mixed-feeding strategies are fuelled by prey type. Most omnivorous dietary guilds appear to have one major evolutionary pathway to a diet similar to their own (e.g. omnivores that specialize on vertebrate prey transitioning to eating both vertebrate and invertebrate prey). There are, however, higher IQRs for some transition rates between mixed-feeding groups, indicating that these transitions are harder to estimate with the current dataset. Despite this uncertainty, our reversible jump MCMC models rarely estimated these transition rates as zero and our constrained MuSSE results also estimated these rates as high. Within omnivory, our models suggest that there are higher rates toward increasing invertebrate specialization and eventually herbivory. Herbivory involves many adaptations diametrically opposed to those for vertebrate prey (e.g. long versus short gut length, flat grinding teeth versus sharp slicing teeth), which would make this dietary transition difficult without intermediate steps using less vertebrate material. For instance, the giant panda, which is estimated to have switched to a mostly bamboo diet approximately 2 Ma [41] still retains the morphologies and the gut microbiome of more omnivorous bear species and has evolved unique ways of dealing with fibrous material that generate lower-quality digestion than other herbivorous species [42–44]. This transition from large omnivore toward greater herbivory highlights the physiological difficulties of moving to drastically new food materials. Overall, our models suggest that transitioning from one specialist group to another requires an intermediate stage of omnivorous or mixed feeding that incorporates both food types. Additionally, transitions out of omnivory into a more specialist diet are probably key moments in evolutionary history and could lead to diversification events, which could explain the clustered phylogenetic signal found for specialist groups like invertivory and herbivory. We infer that these diversification events are more likely to come from invertebrate omnivory, and less often from omnivores that incorporate all food types. This idea agrees with evidence that many early mammalian radiations originated from small insectivorous omnivores [45]. Understanding the difference in ecology and evolution of omnivorous diets could help explain some instances of mammalian diversification and evolution. Potentially, some omnivorous strategies, or lineages, are a stronger evolutionary 'sink' while others may be an evolutionary 'source'.

The existence of two main evolutionary pathways toward increasing invertivory and herbivory and low transition rates into vertivory is most likely driven by the ease of developing physiological and morphological traits that are needed to use the different prey types. The rarity of transitioning into vertebrate prey consumption might be related to the need for certain traits such as increased body size [14] and stronger jaws and teeth [46] in predators that specialize on vertebrate prey. It may also reflect the lower abundance of vertebrate prey relative to insects and plants. For invertivory specifically, higher transitions into invertivorous diets could be because invertebrate prey is abundant and more easily obtained than vertebrate prey. It is also important to note that, while we found high transition rates from vertivory to incorporating more plant and invertebrate material, this trend does not appear to be a common one in the fossil record. Many early mammals are considered to have been invertivorous [45]. In addition, hypercarnivory has been shown to act as an evolutionary ratchet causing hypercarnivores to further specialize on meat consumption, which makes them more vulnerable to extinction [47]. The phylogenetic clustering we found in vertivores is consistent with the idea of such an evolutionary ratchet, although our high rates of transition out of vertivory are not. The reason we find such high transition rates out of vertivory might be related to differences in body mass, as the evolutionary rachet has mostly been found in large hypercarnivores [47] while our vertivore category encompasses a wider range of sizes. The majority of mammals are small bodied, which implies that many of these transitions into invertivory and omnivory are happening in smaller vertivores. This hypothesis would align well with our body mass findings that invertivorous omnivores are smaller than omnivores that incorporate vertebrate prey. An example of such a transition is in the termite specialist the aardwolf, *Proteles cristata*. It is small compared with other extant hyaenids and is thought to have evolved from more vertivorous lineages [48]. Other such transitions can be seen in small feliforms and weasels (figure 2).

## 5. Conclusion

Our findings show that omnivory is not a homogeneous dietary category. Omnivory has different macroevolutionary and macroecological trends that can be hidden when the diet

diversity of omnivores is ignored. Our results show that even for omnivorous mammals, which are phylogenetically over-dispersed, diet, phylogeny and body mass are connected. Despite eating both plants and animals, the body sizes of mammalian omnivores reflect both prey type and evolutionary history. Most omnivorous mammals are small bodied, eat invertebrates and non-fibrous plant material, and are from within Rodentia. Two main evolutionary pathways dominate our transition rate models, one from vertivory to increasingly invertivorous omnivory and ultimately herbivory, and one from vertivory to prey mixing and ultimately invertivory. Therefore, prey type is an under-appreciated but potentially important macroecological variable that future studies of mammalian omnivory should include. Furthermore, the patterns we identified within mammals suggest that studies of dietary evolution within other vertebrate groups could benefit from incorporating prey type to reveal hidden macroevolutionary and ecological trends. Previous work found that omnivorous birds have dissimilar diets to one another, more so than within other diet guilds [8]. This suggests that their diets might be different enough from one another to cause similar ecological and evolutionary trends found in our work. Biologists should consider whether lumping omnivores into the same diet category is ecologically meaningful for the questions being asked, as it may not encapsulate how the diverse array of omnivorous strategies correlate with evolutionary history, evolutionary trends and other ecological traits.

Data accessibility. Data and R code for the analyses are available from the Dryad Digital Repository: https://doi.org/10.5061/dryad.83bk3j9vk [49].

Authors' contributions. D.M.R.: conceptualization, data curation, formal analysis, investigation, methodology, project administration, visualization, writing—original draft, writing—review and editing; S.S.B.H.: funding acquisition, resources, supervision, writing—review and editing; S.A.P.: funding acquisition, resources, supervision, writing—review and editing.

All authors gave final approval for publication and agreed to be held accountable for the work performed herein.

Conflict of interest declaration. We declare no competing interests.

Funding. This study was funded by the National Science Foundation (grant no. DEB-1256897).

Acknowledgements. We would like to thank J. Perry-Houts and J. Crozier for initial code development and the University of Oregon Talapas High Performance Computing cluster team for help with analysis set up on the cluster. K. Smith and L. Roth played an essential role in shaping concepts of dietary categorization in the original dataset. We would like to thank R. Terry, E. Davis, E. Ghezzo, P. Barrett, K. Tate-Jones, H. de Bastos Cruz Machado, A. Peng, D. Grossnickle and three anonymous reviewers for manuscript edits that improved the final version of the manuscript. All silhouettes are from Phylopic.org and are under the Public Domain Mark 1.0 license.

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
