## [Peer Review File · Proceedings of the Royal Society B: Biological Sciences]

Review History

RSPB-2021-1597.R0 (Original submission)

Review form: Reviewer 1 (Facundo Palacio)

Recommendation

Reject – article is not of sufficient interest (we will consider a transfer to another journal)

Scientific importance: Is the manuscript an original and important contribution to its field?

Acceptable

General interest: Is the paper of sufficient general interest?

Acceptable

Quality of the paper: Is the overall quality of the paper suitable?

Good

Is the length of the paper justified?

Yes

Should the paper be seen by a specialist statistical reviewer?

No

Do you have any concerns about statistical analyses in this paper? If so, please specify them explicitly in your report.

No

It is a condition of publication that authors make their supporting data, code and materials available - either as supplementary material or hosted in an external repository. Please rate, if applicable, the supporting data on the following criteria.

Is it accessible?

Yes

Is it clear?

Yes

Is it adequate?

Yes

Do you have any ethical concerns with this paper?

No

Comments to the Author

Dear Prof. Spencer Barrett

Editor-in-Chief, Proceedings of the Royal Society B

First, I would like to mention that I am happy to see a manuscript authored by women only, an unusual phenomenon. Getting straight to the point, this study describes terrestrial mammalian omnivory, how this is distributed across the mammal phylogeny, transition rates between dietary strategies, and how different diet types correlate with body mass. They found that most omnivorous mammals consume only invertebrates and non-fibrous plants, and that invertebrate omnivorous are smaller than vertebrate omnivorous. This challenges the current view of what mammalogists understand by 'omnivory'. What I also found interesting (yet the authors don't explicitly state) is that it shows that using different discrete characterizations of functional groups (here based on diet categories) may be misleading in ecology and evolution, as there can be high variability within such categories. This is not exclusive from mammals, and in my opinion deserves a paragraph in the discussion to gain a broader audience. I know the manuscript has a macroevolutionary focus, but the results presented here can be certainly useful for other ecological subdisciplines (e.g. functional and phylogenetic diversity). Based on these results, I would also expect that authors provide a new classification scheme for omnivores accounting for dietary item and body mass, which will serve as a basis for further mammal dietary studies. My main concern is that the study overall is mainly descriptive (e.g., omnivory description across mammals, phylogenetic signal, transition rates between diet types, correlation between diet and body mass). However, I do believe these kinds of studies are strongly necessary, but I am not fully convinced the work fits the scope of the journal. Please find a few minor comments below.

Title. You refer to mammals, but then analyze 'terrestrial mammals'. I suggest modifying the title, as it may be misleading for the reader.

Ln.22. One key missing aspect is the definition of 'omnivory'. It seems to be related to sentence in ln. 48-49, but a more explicit definition would help the reader to understand what is a mammal omnivore stands for.

Ln. 53. This sentence is a bit confusing. If an animal specializes in a specific food item, is this still an omnivore?

Ln. 59. Does carnivory include insectivorous species? Wouldn't be better to refer to vertebrate predators?

Ln. 71-72. As I said above, not only for evolutionary studies, but also for ecological studies.

Ln. 83. Again, I'm missing what an omnivore represents. From the methods, I infer that omnivores feed on more than one dietary category.

Ln. 84-85. Please include the total number of mammals species to have an idea of species coverage.

Ln. 87. Maybe you can briefly discuss if omnivory in aquatic mammals would suffer from the same issue you expose with terrestrial mammals (i.e., does omnivory in this group deserve to be called into question?).

Ln. 91. The noun 'protein' in this context is uninformative (unless you have information on vertebrate and invertebrate lipids/carbohydrates). Also be consistent in terminology; you refer to 'vertebrate and invertebrate material' below. Finally, add the nature of the data (proportion, presence/absence).

Ln. 108. Using binary categories is less informative than using the proportion of each dietary item (which is understandable not to have this amount of detail for every mammal). So, how does this lower data resolution is expected to impact on your conclusions?

Ln. 162-163. This sentence overlaps with the previous one.

Ln. 185-186. Omit 'with adjusted p-values'. It is clear from the methods.

Ln. 240-243. To support your results, correlations (or the lack of) among dietary categories will also be informative.

Fig. 1. Instead of this figure, you could show the results of the phylogenetic ANOVA, which is more informative.

Fig. 2. It is not clear what the silhouettes correspond to. If they are intended to represent taxonomic groups, maybe you can add vertical bars to clearly distinguish each taxon. Besides, I'm not familiarized with this function, could you describe what do the different colors within a branch represent?

Thank you for letting me reviewing this manuscript and have a good review.

Review form: Reviewer 2

Recommendation

Major revision is needed (please make suggestions in comments)

Scientific importance: Is the manuscript an original and important contribution to its field?

Good

General interest: Is the paper of sufficient general interest?

Good

Quality of the paper: Is the overall quality of the paper suitable?

Good

Is the length of the paper justified?

Yes

Should the paper be seen by a specialist statistical reviewer?

No

Do you have any concerns about statistical analyses in this paper? If so, please specify them explicitly in your report.

No

It is a condition of publication that authors make their supporting data, code and materials available - either as supplementary material or hosted in an external repository. Please rate, if applicable, the supporting data on the following criteria.

Is it accessible?

No

Is it clear?

No

Is it adequate?

No

Do you have any ethical concerns with this paper?

No

Comments to the Author

Review on "What is a mammalian omnivore? Insights into mammalian diet diversity, body mass, and evolution"

This is an important piece of work for ecological studies addressing diet and how we define it. As advocated in previous research, the authors present arguments against simply using "omnivory" to define diet and instead specifying the type of omnivory (specific food types being mixed).

They also evaluate omnivory from an evolutionary standpoint.

I have some suggestions for improving the manuscript, some highlighted below.

My only big concern concerns the studied dataset. With the information provided I could only find a dataset that separates animals between major dietary categories (herbivory, carnivory, and omnivory). The authors mention that this dataset was created after a more detailed collection of data, which I could not find anywhere. More important, the authors need to be more specific about how an animal was determined to feed on a particular food resource from that dataset.

Would feeding on 1% meat and 99% plants make you an omnivore? This is relevant because there are reports of i.e. deer scavenging on carcasses in rare occasions. But who would classify a deer as an omnivore?

I understand that perhaps the original dataset didn't compile percentage utilization data.

However, an alternative would be to work the different dietary categories from more recent published datasets (i.e. Wilman et al. 2014). These datasets detail the actual percentage of each food item that goes in the average diet of a species. This would allow to establish a threshold for what you consider an important food resource, or decide over one with what previous publications have proposed. Additionally, it would allow you to differentiate between i.e. frugivory and granivory, which has been observed to have significant different body masses (see below), and which is relevant for this study.

Introduction

I think you need a paragraph about how other researchers have described/defined omnivory so far, and why this might be problematic. This could either go in the introduction or the discussion. For example, Pineda-Munoz and Alroy (2014) show how using a strict definition of omnivory would lead to classifying the vast majority of mammals into this diet category (Figure 3 in their

manuscript). They also propose a solution for the problem, consisting of listing a main food resource and a secondary food resource (see Table 1 in their manuscript). They identified some of the same omnivore categories you listed.

Additionally, I recommend checking Eisenberg (1981), as they also established some omnivory categories .

Methods

L88: I checked both your SI and the dataset from Price et al. 2012 and I could not find the “more detailed categories and dietary descriptions”. If these were not fully detailed in the original publication, you should provide this data to make sure your analyses are replicable. Similarly, you’ll need to specify the information that went into collecting the dataset. For example, the thresholds (percentage of diet) of each diet resource you followed to determine that an animal is using a particular food resource (Would an animal feeding on 1% fibrous plants and 99% invertebrates be considered to be mixing food resources?)

L90-92: Your “nonfibrous plant parts” category includes “any other plant parts”, which I understand that groups together grains and seeds and fruits. Previous research (Pineda-Munoz et al. 2016) has found significant body mass differences between granivory and frugivory. Since body mass differences are discussed and evaluate here, I think these two specializations should be split. I understand that the dataset where diet resources were extracted from may not differentiate between them, but other really complete datasets such as Elton-Traits do (Wilman et al. 2014). Because Pineda-Munoz et al. (2016) only worked with 139 species I rperformed the analysis with the Elton-Traits database. Differences in log₁₀ body mass using ANOVA between granivores and frugivores were still significant.

L102-107: I don’t fully understand how the diet categories were simplified. Nor how diet categories were treated as binary traits. Please, explain.

L139-145: I like this control analysis, well thought through.

Results:

L163: Both Pineda-Munoz and Alroy (2014) and Wilman et al. (2014) report other species as feeding on fibrous materials and vertebrates (i.e. *Ursus arctos* and *Urocyon cinereoargenteus*). I understand that this might not be true for your dataset, but I think you should at least mention that other datasets disagree on this observation.

L181-182: Again, I wonder how not separating frugivores from granivores might affect your results.

Discussion

L248-249: I suggest rephrasing this sentence

L266-267: This sentence needs rephrasing. You talk about your findings, yet you cite other publications. Mention that your findings agree with previous research for example.

L293-296: Rephrase.

L295-298: As mentioned above, these datasets do exist, and previous research suggests that differences do exist too. So why not using those datasets?

L330-331: Can you look at your data and provide the average body mass and sd of the mammals that transition from carnivory to omnivory or insectivory?

L331-332: What findings are you referring to?

Figures

I really like the figures; they are clear and informative. I only have minor suggestions

Fig1. I personally find log₁₀ of body mass in grams to be easier to interpret when reading a graph. You might consider it.

Fig 2 Caption. Branches are also colored (not just tips) following a specific method for ancestral reconstruction. Mention this briefly.

References

- Eisenberg, J. F. 1981. The mammalian radiations: An analysis of trends in evolution, adaptation, and behavior. University of Chicago Press.
- Pineda-Munoz, S., and J. Alroy. 2014. Dietary characterization of terrestrial mammals. *Proceedings of the Royal Society B: Biological Sciences* 281.

Pineda-Munoz, S., A. R. Evans, and J. Alroy. 2016. The relationship between diet and body mass in terrestrial mammals. *Paleobiology* FirstView:1-11.

Wilman, H., J. Belmaker, J. Simpson, C. de la Rosa, M. M. Rivadeneira, and W. Jetz. 2014. EltonTraits 1.0: Species-level foraging attributes of the world's birds and mammals. *Ecology* 95:2027-2027.

Review form: Reviewer 3

Recommendation

Major revision is needed (please make suggestions in comments)

Scientific importance: Is the manuscript an original and important contribution to its field?

Excellent

General interest: Is the paper of sufficient general interest?

Excellent

Quality of the paper: Is the overall quality of the paper suitable?

Marginal

Is the length of the paper justified?

Yes

Should the paper be seen by a specialist statistical reviewer?

No

Do you have any concerns about statistical analyses in this paper? If so, please specify them explicitly in your report.

Yes

It is a condition of publication that authors make their supporting data, code and materials available - either as supplementary material or hosted in an external repository. Please rate, if applicable, the supporting data on the following criteria.

Is it accessible?

Yes

Is it clear?

Yes

Is it adequate?

Yes

Do you have any ethical concerns with this paper?

No

Comments to the Author

The manuscript by Reuter et al. aims to carefully evaluate what makes a mammal species to be seen as omnivore. The authors compile detailed information on the diets of over 1400 mammal species, and group them according to different criteria. They further analyze different aspects of their biology, such as species richness and body mass distributions, and ultimately estimate transition rates between all the categories.

Overall, the findings not only reinforce what was previously known about the macroevolutionary dynamics of diets in mammals, but add important information on what food types are mostly related to those patterns. I found it especially interesting the results showing that the categorization using the animal component consists in a more solid one, supported by the body size data, when compared to the other categorization using the plant component, with the body size analysis showing expected patterns. I also commend the authors for incorporating phylogenetic uncertainty on the analyses.

However, there are some rather important issues that relate to both some methodological shortcomings as well as to the text as it is written. In my opinion, the text needs clarification throughout, since some of the terminology as well as some convoluted sentences make the text hard to follow in several points. I detail some of those points where I think the text needs some major improvements below.

The main issue is related to the incompleteness of the data used in the rate estimation. I apologize if I got it wrong from the text, but what I could understand is that the authors use in their analyses that use only the species for which they gathered refined dietary data (1437 species). However, to my knowledge BayesTraits does not take into account the undersampling in the data, i.e. it will assume that the combination of phylogeny + traits represent the "full" (phylogenetically speaking) evolutionary history to be analysed. Additionally, the speciation/ extinction dynamics is known to be tightly linked to the transition dynamics, as shown by some of the papers that the authors cite in the text. Hence, by not accounting for neither the undersampling nor the speciation and extinction dynamics, I am not sure how much the results reflect the "true" transition dynamics for the group. I understand that the trait-dependent speciation and extinction models are both "data-hungry" and limited, but I believe that using this approach (even in very limited scenarios, which I suggest ahead) might (or not) alter the results, but would give the authors more realistic estimates of transition rates.

One simple way of using trait-dependent models to get better transition rate estimates would be to constrain the models to have identical speciation and extinction rates among different dietary guilds. This would at least allow the authors to include the sampling percentage into the analysis. Additionally, the authors could also fix the speciation and extinction rates at values that were estimated for each guild by Price et al. 2012 (with the different new omnivore groups assuming the values of Omnivores from Price et al), leaving only the transition rates to be estimated. I fully understand that even in those cases the rate estimates would not be as good as estimates with a more comprehensive sampling, but depending on the distribution of sampled species on the full phylogeny (something I missed from the paper as well) this would at least incorporate part of the limitations of the dataset.

I describe a few minor points below, that I hope can help the authors improve the future versions of the manuscript.

Throughout the whole text: I found that the category names chosen by the authors can be quite confusing in some parts of the text, especially in sentences that compare multiple guilds, so I suggest the authors to try some way to simplify and clarify the names throughout the text.

Line 84: The description of the building of the dataset is confusing. If the data in Price et al (2012) had detailed information, why work with only 1437 species? And if you're working only with the 1437 species, how are they distributed in the tree? Also, I think that if the authors use only those species, the rates of speciation and extinction should matter even more. Lastly, if they only work with these 1437 species, this comprehends not even 1/3 of all mammal species, and thus I fail to see how this would represent well the whole group.

Line 102: Not very clear what was done, maybe improve a little the text.

Line 124: Not very clear what was done, maybe improve a little the text.

Line 131: Text needs to be improved, because on a first read the meaning of the two ranges is not clear. Later in the text it becomes clearer that these are two different parameterizations for the priors.

Line 139: Was this done a single time? If so, I don't think it makes sense to compare this to your empirical results. I can see this as a "null" model for comparison, but this would need a large number of random datasets.

Line 176: Table 3 is quite confusing, as the values don't match. For instance, the range for the log(mass) values are correct, but the mean makes no sense even if the negative is taken out.

Line 189 (Figure 1): Maybe a histogram of frequencies instead of density plots would show this differences more clearly?

Line 213: The SD for the omni-insect is just one order of magnitude lower than the mean, so I'm not sure you can say this

Line 217: I apologize if I'm wrong, but I don't think the absolute likelihood values mean anything. The convergence is what matters, but saying that it converges in lower values do not add anything to the robustness of the results.

Line 221: Out of herb/insect but into other specialized guilds, this should be more clear in the text.

Line 225: There's no figure 4, this should be Figure 3.

Line 299: What do you mean by "evolutionarily constrained"? Also, it is 'dangerous' talking about specialization in this context, because depending on the perspective even a species that eats 100% of invertebrates can be adapted to eating many different types of invertebrates and would thus not be strictly specialized.

Line 309: Evoqueing Carnivora here is a bit out of the blue, and it misleads to a wrong equivalence on carnivores and Carnivora.

Figures: the captions need to be improved, so all three figures are self-contained.

Decision letter (RSPB-2021-1597.R0)

05-Oct-2021

Dear Ms Reuter:

I am writing to inform you that your manuscript RSPB-2021-1597 entitled "What is a mammalian omnivore? Insights into mammalian diet diversity, body mass, and evolution." has, in its current form, been rejected for publication in Proceedings B.

This action has been taken on the advice of referees, who have recommended that substantial revisions are necessary; and the Associate Editor has added some comments. With this in mind we would be happy to consider a resubmission, provided the comments of the referees are fully addressed. However please note that this is not a provisional acceptance.

***Take note: in review it was noted that most data are provided. However, the dataset that went into the analyses is missing. While another publication is listed that has it, the MS mentions that the dataset used is a more detailed version, which cannot be found elsewhere apparently? Please ensure this is fully clear/amended in any resubmission.

Sincerely,
Dr John Hutchinson, Editor
mailto:proceedingsb@royalsociety.org

Associate Editor
Board Member: 1
Comments to Author:

This is an interesting paper that investigates how our definition of omnivory might influence our inferences on macroevolutionary and macroecological inferences. It uses mammals as a studied system to investigate how different types of omnivores compare to each other with respect to body size and how lineages might have transitioned among those different types of omnivores as well as between those and carnivores and herbivores. The paper is mostly well written but there are some passages that need further explanation/justification. The analysis is interesting and well done but there are some important issues pointed out by reviewers (see also some of my comment below) that need to be addressed. Those mostly regard some methodological choices (and potentially new analysis), more info on the data so we can better evaluate it, a discussion or an attempt to deal with the potential effect of species under-sampling, as well as some changes in the narrative to better contrast it with previous work and help to convey its implication beyond mammals. For example, one of the reviewers asks for more information in the data itself, and how species were characterized on those different diet categories. This seems relevant because it touches in the very aspect that the paper is interested in, how our definition of diet categorization might affect our inferences. Also, quite relevant, are the methodological issues raised by one of the reviewers (e.g. not taking into account diversification rates as mentioned by one of the reviewers) and how the results might be influenced data under-sampling. Some of those suggestions include incorporating new analysis (one reviewer point and a point made by myself below), and comparing them to the current analysis. Apart from the very good suggestions given by all reviewers I would add the following (some in fact related to reviewers comments):

1) Given that it is argued that using a simple categorization of omnivory might hide some interesting macroevolutionary and macroecological aspects I think it would be interesting to more directly compare some the results here with the previous results using simpler classifications, in particular those of Price et al 2012 because this comparison might be instructive to not only better understand what was done but also how different categorizations might change or not our interpretations, at the big picture or in more detail. Here some potential ideas:

1.a) How different are the conclusions about omnivory being an evolutionary sink when we consider the new dietary partitioning used in the current paper? I understand the current analysis was not the same as some previous ones (e.g. it did not include the diversification aspect; in fact see comment below) but I think something could be said about this comparison. Taken at face value (and there are, as just mentioned, caveats, e.g. different approaches) the results here also suggest that omnivory (in this case one type of omnivory) is an evolutionary sink right? In a sense the last paragraph (the conclusion) goes in this direction, but I felt more direct contrast between the current results and previous one (e.g. Price et al 2012) might be very informative.

1.b) Related to that, one potentially relevant difference between the current results and those found by Price et al (2012) are that the transition rates (I understand they are measured differently but still) from herbivores to omnivores are by far the highest in Price et al (2012) but are quite small here. Why is that? Is it because the approach was different? If so, does it mean that we need to also include diversification rates in the analysis, as stressed out by one of the reviewers? This should be discussed and justified. In that respect it might be interesting to also evaluate the current diet partitioning using the method used in Price et al (2012) as suggested by one reviewer, or to re-analyze the older dataset/diet partitioning (Price et al, 2012) using the current approach. This will be relevant to evaluate if the differences described above are due to new methods or new categorization, and also relevant for the broader discussion on macroevolutionary sink and how different categorizations might change our general perception.

2) Inferences about rarity of a given dietary category are made and discussed throughout the text, but it might be interesting to also discuss (at least to explicitly explain if the authors do not consider this to be an issue) how differential sampling of different mammal groups (which likely have different proportions of dietary categories) might interfere with this “rarity” assessment, or more broadly with the relative ranks among categories and the analysis in fact. Some of those assessments are very likely to be good (e.g. rarity of omnivores which eat vertebrate + fibrous plant material) but the relative frequencies among other types could be affected by sampling. Nothing is explicitly said about that, but I think it deserves to be.

3) One of the reviewers pointed out that the paper might seem to be too focused on mammal omnivory, but as the reviewer also pointed out, the results/approach discussed here could be also discussed more broadly in terms of classifying other vertebrate species by the different shades of omnivory. I do not think this should require too much ink, but a few sentences on the discussion would make the results more general. In a sense the narrative could benefit, at some passages, from portraying mammals as a model system to study omnivory, and motivate others to do similar investigation in other vertebrate systems.

4) On lines 102 to 107 it is said: “We then used the results of the ANOVA to simplify the dietary guilds to represent only differences in prey type (Table 2). We used these revised diet categories for further analyses to increase our statistical power and decrease our computational time.” Sorry if I missed something (one or two of the reviewers also had trouble understanding this), but I think some information is missing here. Table 2 is about the phylogenetic signal of the newly (I suppose) revisited categories, not the ANOVA result? Was this aforementioned ANOVA done using the categories (or at least some of them) shown on table 1? Sorry I am a bit confused here and this part might need some fixing. I did not fully understand this part so sorry if I am missing something but what is the justification to use body size to create a new categorization of diet. I am clearly missing a step or something here, but others might as well. Might be worth trying to explain this a bit differently.

5) Just out of curiosity. Would the new COMBINE dataset (<https://esajournals.onlinelibrary.wiley.com/doi/full/10.1002/ecy.3344#support-information-section>) or the Elton Traits have a broader coverage in diet information that would allow the analysis to be done with more species? This is an honest question not a rhetoric one. I am not implying this needs to be done but recently came across the publication and wondered about it. It might easily be that those datasets do not have the same needed resolution of data and hence is not amenable to the analysis here. I suspect that is the case so might be worth saying it explicitly in the text.

6) lines 326 to 331 suggests an interesting, and indeed likely, idea. Could it be tested or at least looked even if a bit superficially with the data at hand? e.g. look in the phylogeny for those events and ask whether those occurred from carnivores with lower body size? That might be an interesting addition.

Unfortunately, we cannot accept the paper in its current form. There seems to be quite a few things that need to be done but if the authors think they can properly address those I suggest a new resubmission.

Reviewer(s)' Comments to Author:

Referee: 1

Comments to the Author(s)

Dear Prof. Spencer Barrett

Editor-in-Chief, Proceedings of the Royal Society B

First, I would like to mention that I am happy to see a manuscript authored by women only, an unusual phenomenon. Getting straight to the point, this study describes terrestrial mammalian omnivory, how this is distributed across the mammal phylogeny, transition rates between dietary strategies, and how different diet types correlate with body mass. They found that most omnivorous mammals consume only invertebrates and non-fibrous plants, and that invertebrate omnivorous are smaller than vertebrate omnivorous. This challenges the current view of what mammalogists understand by 'omnivory'. What I also found interesting (yet the authors don't explicitly state) is that it shows that using different discrete characterizations of functional groups (here based on diet categories) may be misleading in ecology and evolution, as there can be high variability within such categories. This is not exclusive from mammals, and in my opinion deserves a paragraph in the discussion to gain a broader audience. I know the manuscript has a macroevolutionary focus, but the results presented here can be certainly useful for other ecological subdisciplines (e.g. functional and phylogenetic diversity). Based on these results, I would also expect that authors provide a new classification scheme for omnivores accounting for dietary item and body mass, which will serve as a basis for further mammal dietary studies. My main concern is that the study overall is mainly descriptive (e.g., omnivory description across mammals, phylogenetic signal, transition rates between diet types, correlation between diet and body mass). However, I do believe these kinds of studies are strongly necessary, but I am not fully convinced the work fits the scope of the journal. Please find a few minor comments below.

Title. You refer to mammals, but then analyze 'terrestrial mammals'. I suggest modifying the title, as it may be misleading for the reader.

Ln.22. One key missing aspect is the definition of 'omnivory'. It seems to be related to sentence in ln. 48-49, but a more explicit definition would help the reader to understand what is a mammal omnivore stands for.

Ln. 53. This sentence is a bit confusing. If an animal specializes in a specific food item, is this still an omnivore?

Ln. 59. Does carnivory include insectivorous species? Wouldn't be better to refer to vertebrate predators?

Ln. 71-72. As I said above, not only for evolutionary studies, but also for ecological studies.

Ln. 83. Again, I'm missing what an omnivore represents. From the methods, I infer that omnivores feed on more than one dietary category.

Ln. 84-85. Please include the total number of mammals species to have an idea of species coverage.

Ln. 87. Maybe you can briefly discuss if omnivory in aquatic mammals would suffer from the same issue you expose with terrestrial mammals (i.e., does omnivory in this group deserve to be called into question?).

Ln. 91. The noun 'protein' in this context is uninformative (unless you have information on vertebrate and invertebrate lipids/carbohydrates). Also be consistent in terminology; you refer to 'vertebrate and invertebrate material' below. Finally, add the nature of the data (proportion, presence/absence).

Ln. 108. Using binary categories is less informative than using the proportion of each dietary item (which is understandable not to have this amount of detail for every mammal). So, how does this lower data resolution is expected to impact on your conclusions?

Ln. 162-163. This sentence overlaps with the previous one.

Ln. 185-186. Omit 'with adjusted p-values'. It is clear from the methods.

Ln. 240-243. To support your results, correlations (or the lack of) among dietary categories will also be informative.

Fig. 1. Instead of this figure, you could show the results of the phylogenetic ANOVA, which is more informative.

Fig. 2. It is not clear what the silhouettes correspond to. If they are intended to represent taxonomic groups, maybe you can add vertical bars to clearly distinguish each taxon. Besides, I'm not familiarized with this function, could you describe what do the different colors within a branch represent?

Thank you for letting me reviewing this manuscript and have a good review.

Referee: 2

Comments to the Author(s)

Review on "What is a mammalian omnivore? Insights into mammalian diet diversity, body mass, and evolution"

This is an important piece of work for ecological studies addressing diet and how we define it. As advocated in previous research, the authors present arguments against simply using "omnivory" to define diet and instead specifying the type of omnivory (specific food types being mixed). They also evaluate omnivory from an evolutionary standpoint.

I have some suggestions for improving the manuscript, some highlighted below.

My only big concern concerns the studied dataset. With the information provided I could only find a dataset that separates animals between major dietary categories (herbivory, carnivory, and omnivory). The authors mention that this dataset was created after a more detailed collection of data, which I could not find anywhere. More important, the authors need to be more specific about how an animal was determined to feed on a particular food resource from that dataset. Would feeding on 1% meat and 99% plants make you an omnivore? This is relevant because there

are reports of i.e. deer scavenging on carcasses in rare occasions. But who would classify a deer as an omnivore?

I understand that perhaps the original dataset didn't compile percentage utilization data. However, an alternative would be to work the different dietary categories from more recent published datasets (i.e. Wilman et al. 2014). These datasets detail the actual percentage of each food item that goes in the average diet of a species. This would allow to establish a threshold for what you consider an important food resource, or decide over one with what previous publications have proposed. Additionally, it would allow you to differentiate between i.e. frugivory and granivory, which has been observed to have significant different body masses (see below), and which is relevant for this study.

Introduction

I think you need a paragraph about how other researchers have described/defined omnivory so far, and why this might be problematic. This could either go in the introduction or the discussion. For example, Pineda-Munoz and Alroy (2014) show how using a strict definition of omnivory would lead to classifying the vast majority of mammals into this diet category (Figure 3 in their manuscript). They also propose a solution for the problem, consisting of listing a main food resource and a secondary food resource (see Table 1 in their manuscript). They identified some of the same omnivore categories you listed.

Additionally, I recommend checking Eisenberg (1981), as they also established some omnivory categories .

Methods

L88: I checked both your SI and the dataset from Price et al. 2012 and I could not find the "more detailed categories and dietary descriptions". If these were not fully detailed in the original publication, you should provide this data to make sure your analyses are replicable. Similarly, you'll need to specify the information that went into collecting the dataset. For example, the thresholds (percentage of diet) of each diet resource you followed to determine that an animal is using a particular food resource (Would an animal feeding on 1% fibrous plants and 99% invertebrates be considered to be mixing food resources?)

L90-92: Your "nonfibrous plant parts" category includes "any other plant parts", which I understand that groups together grains and seeds and fruits. Previous research (Pineda-Munoz et al. 2016) has found significant body mass differences between granivory and frugivory. Since body mass differences are discussed and evaluate here, I think these two specializations should be split. I understand that the dataset where diet resources were extracted from may not differentiate between them, but other really complete datasets such as Elton-Traits do (Wilman et al. 2014). Because Pineda-Munoz et al. (2016) only worked with 139 species I rperformed the analysis with the Elton-Traits database. Differences in log10 body mass using ANOVA between granivores and frugivores were still significant.

L102-107: I don't fully understand how the diet categories were simplified. Nor how diet categories were treated as binary traits. Please, explain.

L139-145: I like this control analysis, well thought through.

Results:

L163: Both Pineda-Munoz and Alroy (2014) and Wilman et al. (2014) report other species as feeding on fibrous materials and vertebrates (i.e. *Ursus arctos* and *Urocyon cinereoargenteus*). I understand that this might not be true for your dataset, but I think you should at least mention that other datasets disagree on this observation.

L181-182: Again, I wonder how not separating frugivores from granivores might affect your results.

Discussion

L248-249: I suggest rephrasing this sentence

L266-267: This sentence needs rephrasing. You talk about your findings, yet you cite other publications. Mention that your findings agree with previous research for example.

L293-296: Rephrase.

L295-298: As mentioned above, these datasets do exist, and previous research suggests that differences do exist too. So why not using those datasets?

L330-331: Can you look at your data and provide the average body mass and sd of the mammals that transition from carnivory to omnivory or insectivory?

L331-332: What findings are you referring to?

Figures

I really like the figures; they are clear and informative. I only have minor suggestions

Fig1. I personally find \log_{10} of body mass in grams to be easier to interpret when reading a graph. You might consider it.

Fig 2 Caption. Branches are also colored (not just tips) following a specific method for ancestral reconstruction. Mention this briefly.

References

Eisenberg, J. F. 1981. *The mammalian radiations: An analysis of trends in evolution, adaptation, and behavior*. University of Chicago Press.

Pineda-Munoz, S., and J. Alroy. 2014. Dietary characterization of terrestrial mammals.

Proceedings of the Royal Society B: Biological Sciences 281.

Pineda-Munoz, S., A. R. Evans, and J. Alroy. 2016. The relationship between diet and body mass in terrestrial mammals. *Paleobiology FirstView*:1-11.

Wilman, H., J. Belmaker, J. Simpson, C. de la Rosa, M. M. Rivadeneira, and W. Jetz. 2014.

EltonTraits 1.0: Species-level foraging attributes of the world's birds and mammals. *Ecology* 95:2027-2027.

Referee: 3

Comments to the Author(s)

The manuscript by Reuter et al. aims to carefully evaluate what makes a mammal species to be seen as omnivore. The authors compile detailed information on the diets of over 1400 mammal species, and group them according to different criteria. They further analyze different aspects of their biology, such as species richness and body mass distributions, and ultimately estimate transition rates between all the categories.

Overall, the findings not only reinforce what was previously known about the macroevolutionary dynamics of diets in mammals, but add important information on what food types are mostly related to those patterns. I found it especially interesting the results showing that the categorization using the animal component consists in a more solid one, supported by the body size data, when compared to the other categorization using the plant component, with the body size analysis showing expected patterns. I also commend the authors for incorporating phylogenetic uncertainty on the analyses.

However, there are some rather important issues that relate to both some methodological shortcomings as well as to the text as it is written. In my opinion, the text needs clarification throughout, since some of the terminology as well as some convoluted sentences make the text hard to follow in several points. I detail some of those points where I think the text needs some major improvements below.

The main issue is related to the incompleteness of the data used in the rate estimation. I apologize if I got it wrong from the text, but what I could understand is that the authors use in their analyses that use only the species for which they gathered refined dietary data (1437 species). However, to my knowledge BayesTraits does not take into account the undersampling in the data, i.e. it will assume that the combination of phylogeny + traits represent the "full" (phylogenetically speaking) evolutionary history to be analysed. Additionally, the speciation/extinction dynamics is known to be tightly linked to the transition dynamics, as shown by some of the papers that the authors cite in the text. Hence, by not accounting for neither the undersampling nor the speciation and extinction dynamics, I am not sure how much the results reflect the "true" transition dynamics for the group. I understand that the trait-dependent speciation and extinction models are both "data-hungry" and limited, but I believe that using this approach (even in very limited scenarios, which I suggest ahead) might (or not) alter the results, but would give the authors more realistic estimates of transition rates.

One simple way of using trait-dependent models to get better transition rate estimates would be to constrain the models to have identical speciation and extinction rates among different dietary guilds. This would at least allow the authors to include the sampling percentage into the analysis. Additionally, the authors could also fix the speciation and extinction rates at values that were estimated for each guild by Price et al. 2012 (with the different new omnivore groups assuming the values of Omnivores from Price et al), leaving only the transition rates to be estimated. I fully understand that even in those cases the rate estimates would not be as good as estimates with a more comprehensive sampling, but depending on the distribution of sampled species on the full phylogeny (something I missed from the paper as well) this would at least incorporate part of the limitations of the dataset.

I describe a few minor points below, that I hope can help the authors improve the future versions of the manuscript.

Throughout the whole text: I found that the category names chosen by the authors can be quite confusing in some parts of the text, especially in sentences that compare multiple guilds, so I suggest the authors to try some way to simplify and clarify the names throughout the text.

Line 84: The description of the building of the dataset is confusing. If the data in Price et al (2012) had detailed information, why work with only 1437 species? And if you're working only with the 1437 species, how are they distributed in the tree? Also, I think that if the authors use only those species, the rates of speciation and extinction should matter even more. Lastly, if they only work with these 1437 species, this comprehends not even 1/3 of all mammal species, and thus I fail to see how this would represent well the whole group.

Line 102: Not very clear what was done, maybe improve a little the text.

Line 124: Not very clear what was done, maybe improve a little the text.

Line 131: Text needs to be improved, because on a first read the meaning of the two ranges is not clear. Later in the text it becomes clearer that these are two different parameterizations for the priors.

Line 139: Was this done a single time? If so, I don't think it makes sense to compare this to your empirical results. I can see this as a "null" model for comparison, but this would need a large number of random datasets.

Line 176: Table 3 is quite confusing, as the values don't match. For instance, the range for the log(mass) values are correct, but the mean makes no sense even if the negative is taken out.

Line 189 (Figure 1): Maybe a histogram of frequencies instead of density plots would show this differences more clearly?

Line 213: The SD for the omni-insect is just one order of magnitude lower than the mean, so I'm not sure you can say this

Line 217: I apologize if I'm wrong, but I don't think the absolute likelihood values mean anything. The convergence is what matters, but saying that it converges in lower values do not add anything to the robustness of the results.

Line 221: Out of herb/insect but into other specialized guilds, this should be more clear in the text.

Line 225: There's no figure 4, this should be Figure 3.

Line 299: What do you mean by "evolutionarily constrained"? Also, it is 'dangerous' talking about specialization in this context, because depending on the perspective even a species that eats 100% of invertebrates can be adapted to eating many different types of invertebrates and would thus not be strictly specialized.

Line 309: Evoqueing Carnivora here is a bit out of the blue, and it misleads to a wrong equivalence on carnivores and Carnivora.

Figures: the captions need to be improved, so all three figures are self-contained.

Author's Response to Decision Letter for (RSPB-2021-1597.R0)

See Appendix A.

RSPB-2022-1062.R0 (Revision)

Review form: Reviewer 3

Recommendation

Accept with minor revision (please list in comments)

Scientific importance: Is the manuscript an original and important contribution to its field?

Excellent

General interest: Is the paper of sufficient general interest?

Excellent

Quality of the paper: Is the overall quality of the paper suitable?

Excellent

Is the length of the paper justified?

Yes

Should the paper be seen by a specialist statistical reviewer?

No

Do you have any concerns about statistical analyses in this paper? If so, please specify them explicitly in your report.

No

It is a condition of publication that authors make their supporting data, code and materials available - either as supplementary material or hosted in an external repository. Please rate, if applicable, the supporting data on the following criteria.

Is it accessible?

Yes

Is it clear?

Yes

Is it adequate?

Yes

Do you have any ethical concerns with this paper?

No

Comments to the Author

In this new version of the manuscript, the authors have addressed virtually all my comments from the previous submission, and I commend them for the thorough work. The manuscript is much improved now after incorporating the suggestions by all reviewers, especially in terms of clarity of the terminology and comparison to analyses that take into account the diversification rates (even though in a scenario with constrained rates, understandably due to limitations in the dataset). It provides important insights on the diversity of ways of being an omnivore in mammals, highlighting the preferential pathways found in the transitions between dietary habits, as well as discussing possible mechanisms that could be also tested in future studies. I think the manuscript constitutes an important piece of research in macroevolution, and I will be happy to see this published. I list below two minor points that might be worth addressing for full clarity.

Line 324: It is a bit ambiguous whether it is small mammals that have capacities to digest fibrous material or that the capacity of mammals to digest fibrous material is small. Based on the following sentences I believe it is the former, but I think this part needs rephrasing for clarity.

Line 362: I don't see this as evidence for low flexibility, but that herbivory and invertivory seem to provide more stability for species (as the authors say right after this sentence). I know this discussion seems semantic, but even though low flexibility and high stability might sound similar, I think that in the context of the paper it is important to discern between the two since (at least for me) "low flexibility" gives the impression of little ability to change, which is very different from being in a stable point because it improves fitness, for instance. Also, the data don't allow the authors to explicitly test this, so I believe the distinction is important.

Review form: Reviewer 4 (David M. Grossnickle)

Recommendation

Major revision is needed (please make suggestions in comments)

Scientific importance: Is the manuscript an original and important contribution to its field?

Good

General interest: Is the paper of sufficient general interest?

Good

Quality of the paper: Is the overall quality of the paper suitable?

Marginal

Is the length of the paper justified?

No

Should the paper be seen by a specialist statistical reviewer?

No

Do you have any concerns about statistical analyses in this paper? If so, please specify them explicitly in your report.

No

It is a condition of publication that authors make their supporting data, code and materials available - either as supplementary material or hosted in an external repository. Please rate, if applicable, the supporting data on the following criteria.

Is it accessible?

Yes

Is it clear?

Yes

Is it adequate?

Yes

Do you have any ethical concerns with this paper?

No

Comments to the Author

See attached file.

Decision letter (RSPB-2022-1062.R0)

16-Aug-2022

Dear Ms Reuter:

Your manuscript has now been peer reviewed and the reviews have been assessed by an Associate Editor. The reviewers' comments (not including confidential comments to the Editor) and the comments from the Associate Editor are included at the end of this email for your reference. As you will see, the reviewers and the Editors have raised some concerns with your manuscript and we would like to invite you to revise your manuscript to address them.

There are numerous important, constructive critiques from the editors and reviewers that deserve careful attention in revision, to avoid rejection, but there is plenty of interest in this study

When submitting your revision please upload a file under "Response to Referees" - in the "File Upload" section. This should document, point by point, how you have responded to the reviewers' and Editors' comments, and the adjustments you have made to the manuscript. We also require a copy of the revised manuscript showing track changes to be uploaded.

Research ethics:

Use of animals and field studies:

It is a condition of publication that data supporting your paper are made available either in the electronic supplementary material. Authors must complete the 'data accessibility' section in the submission system. This should list the database and accession number for all data from the article that has been made publicly available, for instance:

NB. From April 1 2013, peer reviewed articles based on research funded wholly or partly by RCUK must include, if applicable, a statement on how the underlying research materials – such as data, samples or models – can be accessed.

[http://datadryad.org/submit?journalID=RSPB&manu=\(Document not available\)](http://datadryad.org/submit?journalID=RSPB&manu=(Document not available)) which will take you to your unique entry in the Dryad repository. If you have already submitted your data to dryad you can make any necessary revisions to your dataset by following the above link.

Please include the Dryad DOI in the Data Accessibility section and reference in the paper's bibliography.

Please see our Data Sharing Policies (<https://royalsociety.org/journals/authors/author-guidelines/>).

Please submit a copy of your revised paper within three weeks. If we do not hear from you within this time your manuscript will be rejected. If you are unable to meet this deadline please let us know as soon as possible, as we may be able to grant a short extension.

Thank you for submitting your manuscript to *Proceedings B*; we look forward to receiving your revision. If you have any questions at all, please do not hesitate to get in touch.

Best wishes,
Dr John Hutchinson
mailto: proceedingsb@royalsociety.org

Associate Editor

This is a much-improved version of the manuscript. Both reviewers showed enthusiasm for the paper and suggested it might be of interest to a broad audience of ecologists and evolutionary biologists interested in dietary evolution. Although I think the paper has potential and might be of interest to a broad audience, I think there are still several issues that need attention before the paper can be considered for publication. Reviewer 2 has listed several important issues that when addressed should greatly improve the manuscript. Apart from those I would add the following:

1) Summarize the results from the MuSSE model, in particular the transition rates which is the focus of the paper, into a figure. In its current form (a table in the supplemental material) it is very difficult to compare it to the main results (for example compare figure 3 to supplemental table 7) or the unconstrained and constrained MuSSE results among themselves. In fact a simple scatter plot for the unconstrained and constrained MuSSE results suggest they are quite different.....

2) Given that one of the main points is to suggest that we gain ecological and evolutionary insights when looking at omnivory in more detail, I think a figure comparing the main results when sub-dividing omnivory into further categories to a scenario where we have only 3 categories (as in Price et al) would be very informative. For example, a figure comparing the transition rates when using all categories to 3 categories. That could be done to show the transition rates from all different analysis (see comment below) from the 2 scenarios as well. Given space limits, one possibility would be to add such figure and move some of those tables to the supplemental material. Alternatively, a panel could be added to figure 3, or a the transition rate figure comparing all methods and scenarios could be a supplemental figure. My point is that several of the arguments rely on comparing the transition rates either among different methods (reversible jump MCMC vs MuSSE), or between scenarios with multiple omnivorous categories to a scenario with just one omnivory category, hence a figure would be helpful.

3) Sorry to just notice this now, but I was wondering why the authors decided to use the Faurby & Svenning (2015) phylogeny instead of a more recent one by Upham and colleagues. Would there be any specific reason why? Do you expect the (small???) differences in topology and/or taxonomy to be relevant?

4) Although it is not the focus of the paper, I think you could present the diversification MuSSE results for the unconstrained analysis in a supplemental material figure. It would help the narrative at some parts to mention those results and it might be instructive to see those values (e.g. lines 364-372; or lines 379-382).

Reviewer(s)' Comments to Author:

Referee: 3

Comments to the Author(s).

In this new version of the manuscript, the authors have addressed virtually all my comments from the previous submission, and I commend them for the thorough work. The manuscript is much improved now after incorporating the suggestions by all reviewers, especially in terms of clarity

of the terminology and comparison to analyses that take into account the diversification rates (even though in a scenario with constrained rates, understandably due to limitations in the dataset). It provides important insights on the diversity of ways of being an omnivore in mammals, highlighting the preferential pathways found in the transitions between dietary habits, as well as discussing possible mechanisms that could be also tested in future studies. I think the manuscript constitutes an important piece of research in macroevolution, and I will be happy to see this published. I list below two minor points that might be worth addressing for full clarity.

Line 324: It is a bit ambiguous whether it is small mammals that have capacities to digest fibrous material or that the capacity of mammals to digest fibrous material is small. Based on the following sentences I believe it is the former, but I think this part needs rephrasing for clarity.

Line 362: I don't see this as evidence for low flexibility, but that herbivory and invertivory seem to provide more stability for species (as the authors say right after this sentence). I know this discussion seems semantic, but even though low flexibility and high stability might sound similar, I think that in the context of the paper it is important to discern between the two since (at least for me) "low flexibility" gives the impression of little ability to change, which is very different from being in a stable point because it improves fitness, for instance. Also, the data don't allow the authors to explicitly test this, so I believe the distinction is important.

Referee: 4

Comments to the Author(s).

See attached file.

Author's Response to Decision Letter for (RSPB-2022-1062.R0)

See Appendix B.

RSPB-2022-1062.R1

Review form: Reviewer 4 (David M. Grossnickle)

Recommendation

Accept with minor revision (please list in comments)

Scientific importance: Is the manuscript an original and important contribution to its field?

Good

General interest: Is the paper of sufficient general interest?

Good

Quality of the paper: Is the overall quality of the paper suitable?

Good

Is the length of the paper justified?

Yes

Should the paper be seen by a specialist statistical reviewer?

No

Do you have any concerns about statistical analyses in this paper? If so, please specify them explicitly in your report.

No

It is a condition of publication that authors make their supporting data, code and materials available - either as supplementary material or hosted in an external repository. Please rate, if applicable, the supporting data on the following criteria.

Is it accessible?

Yes

Is it clear?

Yes

Is it adequate?

Yes

Do you have any ethical concerns with this paper?

No

Comments to the Author

Thank you for making edits to address most of my previous comments. I don't have any remaining major concerns with the paper. My line comments below are mostly suggested edits for clarity. However, I do reiterate below a couple of points from my last review that I don't think you addressed in your revisions (5% cutoff for diet choices, justification for merging herbivore groups) - I recommend adding some text to your paper to help clarify these points.

Congrats on the nice study!

Dave Grossnickle
dmgrossn@uw.edu

Lines 18-19: "We find consuming all four food types is relatively rare, ..." is a little vague. And it's different than what you have in your marked manuscript draft, which is "We find that omnivores that consume all four food types are relatively rare, ...". I recommend using the latter version.

Line 25-27 (and similarly Lines 464-465): "Our work highlights that omnivorous mammals are diverse, have different evolutionary origins, and consume different foods." These three conclusions are all fairly obvious and/or vague. E.g. from a phylogenetic standpoint, any diet that evolves more than once has "different evolutionary origins," so that phrase doesn't really tell us anything. I recommend revising the sentence to include more specific conclusions, or have a sentence that gives recommendations to other researchers (e.g., something like your concluding sentence on Lines 480-482).

Line 40: I'd expand "omnivores" to "omnivores on average" or something similar, because there is a lot of overlap among body sizes and tooth morphologies of diet groups, so not all omnivores are between carnivores and herbivores.

Lines 84-113: Your edits to this paragraph are helpful, but I still found it a bit confusing and backwards. Before you even mention your methods, you first focus on what you did NOT do (Lines 86-95) - you exclude aquatic mammals, didn't base diets on primary food choices (sensu Pineda-Munoz et al.), and didn't analyze proportional data because it's unreliable for some taxa. (The last point is especially confusing because you later say that your diet categories are based on

proportional data, so you are indirectly using proportional data.) It reads as very defensive/negative. I recommend revising the paragraph so that you start by stating your key methods (e.g. you could move up Lines 95-96 and 106-111 to the start of the paragraph). And then at the end of the paragraph/subsection you could explain why you don't include aquatic mammals and prefer your dataset over the larger EltonTraits dataset.

Line 99: "fewer species" without a subsequent "than" statement is unclear. I assume based on earlier text that you're referring to the EltonTraits dataset, but I recommend stating that. Or you could erase or revise this text, especially if you follow my above suggestion to revise this paragraph -- I don't think you need to be as defensive about not using EltonTraits.

Line 109: Maybe I missed it, but I don't think you responded to this comment in my previous review: "Why 5%? If there's a reason, you should note it. If it's arbitrary, then you could run sensitivity analyses with different cutoff percentages to see how your choice influences analyses like pANOVAs (e.g. I did this as supp analyses in Grossnickle 2020). If you choose not to follow that suggestion (which is fine - it might be more work than it's worth), you should at least acknowledge that this choice may influence results. For instance, if you instead used 10%, then you'd probably have fewer species categorized as generalists, and more classified as carnivores and herbivores. This might influence downstream analyses."

I understand if you don't run any new analyses, but I do think that your choice of 5% could have a big impact on results, and I recommend at least briefly acknowledging this in your paper.

Line 120: "implemented in the caper package". I'd expand this to something like "implemented by using functions in the caper package".

Lines 135-144: I appreciate that you added a couple of sentences to justify your simplification of diet groups based on omnivore body mass patterns. But 1) you still don't even mention in this paragraph that you merged herbivore groups, which is an important decision that should be explained in the text (unless you already explain it elsewhere and I missed it), and 2) you aren't clear on your justification for merging herbivore groups - your current text (especially the first sentence, Lines 135-137) implied that all merging decisions (omnivores and herbivores) were based on omnivore body mass patterns, which doesn't make any sense for herbivores because they have their own body mass patterns. You mention computational limits and comparisons to Price et al. (2012) at the end of the paragraph, but because at this point you've only mentioned omnivores, it sounds like those are just additional reasons for merging omnivore groups. I recommend explicitly stating that those are reasons for merging herbivore groups.

Line 138: To help clarify the goal of the sentence, maybe expand "Body mass is directly ..." to something like "We chose to use body mass patterns as a determining factor for merging groups because body mass is directly ...".

Line 341: I'm a little confused as to what you mean by "tightly evolutionarily constrained," as well as confused by how the results you cite support that conclusion. I recommend adding some more details here to clarify your point.

Lines 464-465: See my comments above on Lines 25-27. The three points listed in this sentence are so vague that they don't tell the reader anything.

Decision letter (RSPB-2022-1062.R1)

31-Oct-2022

Dear Ms Reuter

I am pleased to inform you that your manuscript RSPB-2022-1062.R1 entitled "What is a mammalian omnivore? Insights into terrestrial mammalian diet diversity, body mass, and evolution." has been accepted for publication in *Proceedings B*.

The referee(s) have recommended publication, but also suggest some minor revisions to your manuscript. Therefore, I invite you to respond to the referee(s)' comments and revise your manuscript. Because the schedule for publication is very tight, it is a condition of publication that you submit the revised version of your manuscript within 7 days. If you do not think you will be able to meet this date please let us know.

Here, there are simply some changes to the main text needed, especially to justify some assumptions.

[http://datadryad.org/submit?journalID=RSPB&manu=\(Document not available\)](http://datadryad.org/submit?journalID=RSPB&manu=(Document%20not%20available)) which will take you to your unique entry in the Dryad repository. If you have already submitted your data to dryad you can make any necessary revisions to your dataset by following the above link. Please see <https://royalsocietypublishing.org/journal/rsos/ethics-policies/data-sharing-mining/> for more details.

6) For more information on our Licence to Publish, Open Access, Cover images and Media summaries, please visit <https://royalsocietypublishing.org/journal/rsos/authors/author-guidelines/>.

Sincerely,
Dr John Hutchinson
Editor, Proceedings B
<mailto:proceedingsb@royalsocietypublishing.org>

Reviewer(s)' Comments to Author:

Referee: 4

Comments to the Author(s)

Thank you for making edits to address most of my previous comments. I don't have any remaining major concerns with the paper. My line comments below are mostly suggested edits for clarity. However, I do reiterate below a couple of points from my last review that I don't think you addressed in your revisions (5% cutoff for diet choices, justification for merging herbivore groups) – I recommend adding some text to your paper to help clarify these points.

Congrats on the nice study!

Dave Grossnickle
dmgrossn@uw.edu

Lines 18-19: "We find consuming all four food types is relatively rare, ..." is a little vague. And it's different than what you have in your marked manuscript draft, which is "We find that omnivores that consume all four food types are relatively rare, ...". I recommend using the latter version.

Line 25-27 (and similarly Lines 464-465): "Our work highlights that omnivorous mammals are diverse, have different evolutionary origins, and consume different foods." These three conclusions are all fairly obvious and/or vague. E.g. from a phylogenetic standpoint, any diet

that evolves more than once has “different evolutionary origins,” so that phrase doesn’t really tell us anything. I recommend revising the sentence to include more specific conclusions, or have a sentence that gives recommendations to other researchers (e.g., something like your concluding sentence on Lines 480-482).

Line 40: I’d expand “omnivores” to “omnivores on average” or something similar, because there is a lot of overlap among body sizes and tooth morphologies of diet groups, so not all omnivores are between carnivores and herbivores.

Lines 84-113: Your edits to this paragraph are helpful, but I still found it a bit confusing and backwards. Before you even mention your methods, you first focus on what you did NOT do (Lines 86-95) – you exclude aquatic mammals, didn’t base diets on primary food choices (sensu Pineda-Munoz et al.), and didn’t analyze proportional data because it’s unreliable for some taxa. (The last point is especially confusing because you later say that your diet categories are based on proportional data, so you are indirectly using proportional data.) It reads as very defensive/negative. I recommend revising the paragraph so that you start by stating your key methods (e.g. you could move up Lines 95-96 and 106-111 to the start of the paragraph). And then at the end of the paragraph/subsection you could explain why you don’t include aquatic mammals and prefer your dataset over the larger EltonTraits dataset.

Line 99: “fewer species” without a subsequent “than” statement is unclear. I assume based on earlier text that you’re referring to the EltonTraits dataset, but I recommend stating that. Or you could erase or revise this text, especially if you follow my above suggestion to revise this paragraph -- I don't think you need to be as defensive about not using EltonTraits.

Line 109: Maybe I missed it, but I don’t think you responded to this comment in my previous review: “Why 5%? If there’s a reason, you should note it. If it’s arbitrary, then you could run sensitivity analyses with different cutoff percentages to see how your choice influences analyses like pANOVAs (e.g. I did this as supp analyses in Grossnickle 2020). If you choose not to follow that suggestion (which is fine – it might be more work than it’s worth), you should at least acknowledge that this choice may influence results. For instance, if you instead used 10%, then you’d probably have fewer species categorized as generalists, and more classified as carnivores and herbivores. This might influence downstream analyses.”

I understand if you don’t run any new analyses, but I do think that your choice of 5% could have a big impact on results, and I recommend at least briefly acknowledging this in your paper.

Line 120: “implemented in the caper package”. I’d expand this to something like “implemented by using functions in the caper package”.

Lines 135-144: I appreciate that you added a couple of sentences to justify your simplification of diet groups based on omnivore body mass patterns. But 1) you still don’t even mention in this paragraph that you merged herbivore groups, which is an important decision that should be explained in the text (unless you already explain it elsewhere and I missed it), and 2) you aren’t clear on your justification for merging herbivore groups – your current text (especially the first sentence, Lines 135-137) implied that all merging decisions (omnivores and herbivores) were based on omnivore body mass patterns, which doesn’t make any sense for herbivores because they have their own body mass patterns. You mention computational limits and comparisons to Price et al. (2012) at the end of the paragraph, but because at this point you’ve only mentioned omnivores, it sounds like those are just additional reasons for merging omnivore groups. I recommend explicitly stating that those are reasons for merging herbivore groups.

Line 138: To help clarify the goal of the sentence, maybe expand “Body mass is directly ...” to something like “We chose to use body mass patterns as a determining factor for merging groups because body mass is directly ...”.

Line 341: I'm a little confused as to what you mean by "tightly evolutionarily constrained," as well as confused by how the results you cite support that conclusion. I recommend adding some more details here to clarify your point.

Lines 464-465: See my comments above on Lines 25-27. The three points listed in this sentence are so vague that they don't tell the reader anything.

Decision letter (RSPB-2022-1062.R2)

14-Nov-2022

Dear Ms Reuter

I am pleased to inform you that your manuscript entitled "What is a mammalian omnivore? Insights into terrestrial mammalian diet diversity, body mass, and evolution." has been accepted for publication in Proceedings B.

You can expect to receive a proof of your article from our Production office in due course, please check your spam filter if you do not receive it. PLEASE NOTE: you will be given the exact page length of your paper which may be different from the estimation from Editorial and you may be asked to reduce your paper if it goes over the 12 page limit.

If you are likely to be away from e-mail contact during this period, let us know. Due to rapid publication and an extremely tight schedule, if comments are not received, we may publish the paper as it stands.

Your article has been estimated as being 11 pages long. Our Production Office will be able to confirm the exact length at proof stage.

Data Accessibility section

Open access

The open access fee is £1,700 per article (plus VAT for authors within the EU). Payment of open access fees will enable your article to be made freely available via the Royal Society website as soon as it is ready for publication. For more information about open access publishing please visit our website at http://royalsocietypublishing.org/site/authors/open_access.xhtml. If you have opted for Open Access in Proceedings B, payment of an article processing charge (APC) may be due before your article is published. Our partner Copyright Clearance Center's RightsLink for Scientific Communications will contact the corresponding author about your open access options from the email domain @copyright.com (if you have any queries regarding fees, please see <https://royalsocietypublishing.org/rspb/for-authors#question12> or contact authorfees@royalsociety.org). If you now wish to opt for open access then please let us know as soon as possible.

Page charges (for non-Open Access papers)

Our partner Copyright Clearance Center's RightsLink for Scientific Communications will contact the corresponding author about payment for page charges.

Sincerely,
Proceedings B
mailto: proceedingsb@royalsociety.org

Appendix A

Response to Reviews

May 2022

Dear Editor,

Please consider our revised manuscript entitled *What is a mammalian omnivore? Insights into terrestrial mammalian diet diversity, body mass, and evolution* for publication in *Proceedings B*. We appreciated the feedback we were given on our review in October of 2021. Many of the issues raised by both you, the Editor, and the reviewers were insightful and gave us ways to move forward in improving this manuscript. We addressed many of the concerns about the analysis, writing, and discussion and we think the manuscript is now a stronger scientific work.

The reviews brought up major issues that we found appropriate and fixed in the new version of the manuscript:

1. We agreed with reviewers that our terminology was not consistent throughout the paper. We updated terminology throughout the paper to make the diet categories clearer and less confusing. We replaced the confusing terms ‘vertebrate prey specialists’ and ‘predators that specialize on vertebrate prey’ with the more concise terms ‘vertivorous’, ‘vertivory’ and ‘vertivores’. We also changed our use of insectivore to invertivore to better reflect the data we used.

Changing these terms also allowed for the omnivore categories to become easier to read in sentences comparing our results. For instance ‘omnivores specializing on both invertebrate and vertebrate prey’ now reads as ‘invertivorous/vertivorous omnivory’.

We changed the terms in the tables and figures to follow these new consistent terms. Additionally, we also added a column to table 1 that includes the trophic level of the dietary categories to help show which categories are omnivorous categories. We hope this makes our categories clearer and easier to read throughout the paper.

2. Concerns that our dataset and results should directly compare to the results published by Price et al. 2012 were addressed in a number of ways. We first ran a BayesTraits analysis identical to our own using just the three trophic categories used by Price et al. We have now added a description of this analysis to the methods, results, and discussion. This strengthened the paper and allowed us to expand on our concerns of lumping omnivores into one category.

We next ran a MuSSE analysis similar to those conducted by Price et al. 2012 with our new omnivore diet categories that separate out prey type. We incorporated the sampling frequency to be 0.24 of all Mammalia. We first did this on ten trees with the extinction and speciation constrained to the rates found in Price et al. 2012. These results agreed with our BayesTraits rates. We then ran another round of MuSSE using the same ten trees but had everything unconstrained. The models performed less well, and the spread of data was quite high for many of the transition rates but they also had similar results to those we found using BayesTraits.

We think that our discussion now clearly uses the results of all three models to make the case that using the finer diet categories resulted in a better picture of the evolutionary origins of different kinds of omnivory.

3. Concerns over our dataset brought up by Reviewer 2 were addressed by adding more details about the dataset in the methods section, specifically lines 88-103. Hopefully these clear up any confusion about our definition for omnivory and how our data differs from other published datasets. We also included a sentence that explains how our dataset differs from more complete but lower resolution datasets, such as Elton Traits. Additionally, our detailed diet dataset, and where to find the references have been added to the supplemental material.
4. There were concerns that our rarity assessment of diets could be affected by sampling bias. We looked at the number of mammals in the IUCN and compared our dataset to the numbers reported for each order. We put this data in the supplemental. We believe that we have fairly good coverage for many of the mammalian orders. As mentioned earlier we also incorporated the sampling frequency to be 0.24 of all Mammalia in our MuSSE model and did not see a change in the pattern of transition rates. So we believe that despite only having about $\frac{1}{3}$ of mammals we have a good taxonomic sampling. We also added the phrase “our dataset” in front of many of our interpretations of diversity. This can be seen in lines 278 and 285.
5. It was also pointed out that we should broaden our work to apply to groups other than mammals. Last paragraph in discussion was therefore modified to suggest that our findings could apply to other groups such as birds.

Minor comments:

We would like to thank you, the Editor, for many of the insightful comments

We would like to thank the Editor for pointing out the new COMBINE dataset. We did not know of its existence and it became useful for a separate project. We decided against using it because it does not contain the data resolution that we were aiming for. We know that many datasets exist that have broader coverage than the dataset that we used but many of them use dietary descriptions from secondary sources as opposed to primary references. We hope our addition to Ln 88-103 clears up our decision to use the Price et al. 2012 dataset. Additionally, as mentioned earlier, using the Price et al. 2012 dataset allowed us to make direct comparisons with those past results which we believed strengthens the paper’s main argument.

Ln 120-121 Unclear language in the original version made it confusing to understand how the diet categories were simplified. We added a description of what to look for in Table 2.

Ln 424-425 The editor suggested we add a bit more about if transitions out of vertivory are happening in small vertivorous lineages. We looked at our ancestral state reconstructions in Figure 2 and noted that indeed many of these transitions happen in small feliforms and weasels.

Reviewer 1 gave a few insightful minor edits. We have addressed many of them with the following edits:

Title: terrestrial was added to the title to make it clear what group we focused on.

Ln 48 a definition of omnivory has been added.

Ln 83 “about a quarter of all mammal species” has been added to the methods.

Ln 106 protein has been replaced with prey. We did this throughout the paper as well.

Ln 215 Removed “with adjusted p-values” as suggested.

Figure 1. We thought the density plots still provided a clear image of the pattern captured in the ANOVA results so we kept it as is.

Figure 2. We updated the figure to have the major mammalian groups denoted with bars and labels. Doing this also made the silhouettes more clear. We also updated the figure caption to better describe the patterns and the model used.

Reviewer 2 also gave insightful comments about the dataset and language used. We have addressed many of them with the above major edits and the following minor edits:

Ln 120-121 Unclear language in the original version made it confusing to understand how the diet categories were simplified. We added a description of what to look for in Table 2.

Ln 188 Unclear language in the original version made our description confusing. We added the word “only” to reflect that we were talking about mammals that solely consume fibrous plant material and vertebrate material.

Ln 421-422 We added “that invertivorous omnivores are smaller than omnivores that incorporate vertebrate prey” to make it clear what findings we were referring to.

Figure 2 caption was updated to explain how the ancestral states on the branches were reconstructed.

Reviewer 3 also gave insightful comments about the analyses used and the language used in interpreting our results. We have addressed many of them with the above major edits and the following minor edits:

As mentioned earlier category names were simplified and improved throughout the paper.

Ln 88-103 we added more detail about the collection of the dataset and coverage to the methods and supplementary data so hopefully this clears up any confusion about the dataset. We also believe our MuSSE models confirmed that our rates are meaningful despite having only $\frac{1}{3}$ of mammals.

Ln 148-150 We now use a new sentence to describe the hyperpriors used in our RJ MCMC analysis “We used uniform hyperpriors to seed the exponential prior on the parameters, seeding the mean of the exponential prior from a uniform distribution on the interval 0 to 10 or 0 to 2.” Hopefully this more clearly describes how we ran two clusters of models with two hyperprior distributions.

Ln 156-158 Reviewer 3 thought that we would need a large number of random datasets to compare to our results. We believe this is unnecessary for a few reasons. The random dataset we did run was very inconsistent with its model runs and each run had a hard time converging, suggesting that we did in fact erase the evolutionary structure of our diet data. The runs were also extremely different from the true dataset. If it was closer we would be concerned but it was so different that we can be confident with our comparison between it and our dataset. The random showed that the evolutionary structure is what matters here. Each run takes approximately 30 days and we believe it would be a long wait for little return. We did, however, remove the word “significant” and replaced it with “potentially meaningful” when explaining our reasoning for the analysis.

Ln 240 We removed the sentence about low SD showing that sample size is unlikely influencing our estimates and instead just stated that the SD is low for those groups.

Ln 245-246 Reviewer 3 mentioned that the absolute likelihood values probably can't be compared. We removed “converged on lower likelihood values and instead added “overall higher median transition rates and higher IQRs”.

Ln 363-365 Reviewer 3 had a very good insight about how mentioning that specialist groups are “evolutionarily constrained” was ignoring that many specialists can be generalists within their own diet type. We instead modified our sentence to say “The combined patterns of high diversity, high phylogenetic clustering, and low transition rates into other diet types suggests that herbivory and invertivory are highly successful and evolutionarily stable strategies” We hope this clarifies that we were trying to show that they have few changes on the mammalian tree.

Ln 377 we removed the sentence “This result should come as no surprise as the order Carnivora is known for its diversity of diets”. We agreed with reviewer 3 that it does not add much to our argument and suggests that all vertivores are within Carnivora.

Figure captions have all been improved to help the figures stand alone.

Table 3. There was some confusion about the number and how the ranges do not match. Please correct us if we are wrong but we believe these numbers to be correctly reported.

For even more detail on how we edited our manuscript please see the word document with track changes. If there are questions or concerns please let us know.

This manuscript has not been published elsewhere and is not under consideration by another journal. All authors have approved the manuscript and agree with its submission.

Thank you for your consideration,

Dana M. Reuter*, Samantha S.B. Hopkins, Samantha A. Price

Dana M. Reuter
Department of Earth Sciences
1272 University of Oregon, Eugene, OR 97403, USA
dreuter@uoregon.edu

Appendix B

Dear Editor,

Please consider our revised manuscript entitled *What is a mammalian omnivore? Insights into terrestrial mammalian diet diversity, body mass, and evolution* for publication in *Proceedings B*. We appreciated the feedback we were given on our review, it allowed us to greatly improve this manuscript. We addressed many of the concerns about the analysis, writing, and discussion and we think the manuscript is now a stronger scientific work.

The reviews brought up major issues that we found appropriate and fixed in the new version of the manuscript:

1. A new figure that summarizes all the model results has replaced Figure 3. Figure 3 now has the MuSSE results summarized and the transitions between the three trophic categories summarized to make it easier to compare all the results.
2. Reviewer 4 had concerns that we did not discuss the work of Pineda-Munoz & Alroy 2014, and Pineda-Munoz & Alroy 2016 enough, especially when we refer to “generalists”. We updated many of the comments about generalists to be more descriptive, so that the concept would flow better with our results and discussion. We also add more to the discussion to highlight how our work supports or disagrees with the findings of Pineda-Munoz & Alroy 2014 (Ln 306-311 and Ln 365-366).
3. Reviewer 4 brought up some concerns that our justification for our simplified diet groups as not strong enough. We added a few sentences to the methods (Ln 137-146) to explain our reasons for the simplified the diet groupings based on mass. Firstly, because body mass is an important trait that is related to the energy requirements of organisms and it co-varies with other traits. Secondly, we explain that the groupings, including grouping the herbivores, allowed us to compare to and add to the results of Price et al. 2012 while still reducing our computational time.
4. Reviewer 4 was concerned that we did not compare our results to those of Price et al. 2012 and we did not revisit the idea of omnivory being an evolutionary sink in our discussion despite bringing it up in the introduction. We agree that this an important point that was overlooked in our previous version. We have restructured our discussion (Ln 376-393, Ln 409-411, and Ln434-440) to make the comparison between our work and Price et al. 2012 stronger. We state that differences between the model parameters are most likely responsible for the differences in herbivore-to-omnivore transition rates. We also state “while herbivorous lineages may lead to many omnivorous species, as was suggested by Price et at. 2012, they are constrained by prey type and potentially small bodied.” We think this is an important insight and addition that shows how our paper adds to this discussion.
5. Reviewer 4 pointed out that our sampling frequency should be updated, and we thank them for referring us to mammaldiversity.org, it was very helpful as previously we were

using numbers from the IUCN. We reran our MuSSE models with the new sampling frequency (which did not change the outcome) and updated the figures.

6. It was suggested by Reviewer 4 to move results text and tables to a Supplemental results file to shorten the paper. We moved two tables to a supplementary file. We also rearranged the supplemental files work with the Figshare and Dryad platforms. All datafiles are now stored in Dryad.
7. Reviewer 4 suggested that we update our ANOVA methodology. We thank them for this suggestion and having now read Collyer & Adams 2022 and discussing the issues with Mike Collyer we agree. We have re-run all our phylogenetic ANOVA's using a PGLS ANOVA implemented the caper package. We have updated these results in our paper (Ln 222-238). We also add more to our discussion about how this information shows that although omnivorous diets have low phylogenetic signal, there are very large differences in omnivore ecology that exists among the orders of mammals (Ln 338-345). We believe this insight makes our paper more accurately represent the complexities that can arise between ecology and evolution. Our paper now suggests that not only is prey type a potentially important dietary variable, but the evolutionary origins of omnivorous animals is also important. This change prompted by Reviewer 4 has greatly improved the paper.

Major comments that were not addressed in this manuscript:

Regarding the use of the more recent Upham phylogeny, one of the reasons we did not use it in our paper is because we had already started running our models by the time it was published. The analyses take long time to complete because of the number of diet categories, around 30 days for each chain. We do not believe that using the more recent tree would substantially change the results of our study, as the higher-level relationships are relatively stable between studies and most of our patterns appear to be driven by processes operating at these larger scales.

Minor comments:

Reviewer three pointed out two places where our wording was causing some confusion. We therefore updated our sentences to clarify our meaning.

Ln 354 We added text to make it clear that we were referring to the capacities to digest fibrous material by small mammals.

Ln 401-402 We agreed with the comment that there is a subtle difference between stability and flexibility and that our study does not explicitly test this. We removed the mention of low dietary flexibility.

Reviewer 4 made many insightful comments. We have addressed many of them with the following minor edits:

We updated all references to the tree of life to be instead, "phylogeny" throughout the paper.

Ln 101 We updated our description of the Price et al. 2012 dataset to say that it is categorical but was based on studies that reported some proportional data. We refrain from describing the dataset in too much detail because this was already done in Price et al. 2012.

Ln 104 We deleted the extra reference to previously published data.

Ln 164 We defined IQR so that the acronym is defined for the rest of the paper.

Ln 179 We stated that we did not perform a Bayesian MCMC analysis because of computational time constraints.

Ln 188-207 Changed references to dataset, to sample.

Ln 222 We incorporated the Levene's test results into the text and have deleted the supplementary table.

Ln 268 Changed the wording from "seven categories identified by our ANOVA" to "seven simplified categories" because the ANOVA only was used on the omnivorous groups.

We agreed that our earlier statements about prey type being an important driving factor were a bit too strong given our study set up. We followed Reviewer 4's suggestion and changed those sentences throughout the paper to have more words like "hypothesize" and "our results suggest". Examples can be found on Ln 296 and Ln 470.

Tables and figures have improved captions to improve their readability.

Figure 2 has an improved caption that communicates that it is not an accurate representation of the reconstructed ancestral states or transition rates. We also replaced the simmap with one that more closely resembles early mammal evolution.

All images from Phylopic were selected because they were under the Public Domain Mark 1.0 license. This has been added to the Acknowledgments section.

For even more detail on how we edited our manuscript please see the word document with track changes. If there are questions or concerns, please let us know.

This manuscript has not been published elsewhere and is not under consideration by another journal. All authors have approved the manuscript and agree with its submission.

Thank you for your consideration,

Dana M. Reuter*, Samantha S.B. Hopkins, Samantha A. Price

Dana M. Reuter
Department of Earth Sciences
1272 University of Oregon, Eugene, OR 97403, USA
dreuter@uoregon.edu

Appendix C

Dear Editor,

Please see our revised manuscript entitled *What is a mammalian omnivore? Insights into terrestrial mammalian diet diversity, body mass, and evolution* for publication in *Proceedings B*. We addressed the concerns about clarity in our writing, and methods and we think the manuscript is now a stronger scientific work.

The review brought up issues that we found appropriate and fixed in the new version of the manuscript:

We agreed with comments on Ln 18-19 and changed the sentence to say: “We find omnivores that consume all four food types are relatively rare, as most omnivores consume only invertebrate prey and non-fibrous plants.”

We edited the last sentence of the abstract to better reflect our main takeaways. It now says: “Future work should avoid lumping omnivores into one category given the ecological variety of omnivore diets and their strong evolutionary influence.”

We edited Ln 40 from “we have learned that omnivores” to “we have learned that omnivores on average”. We agree that this makes our description of the trends align better with reality.

Ln 84-110 We agreed that the paragraph needed to be revisited. It had lost its clarity with all the additions. We tried condensing the paragraph while still keeping the main points about how the dataset was compiled and why we chose to use it. We think it is now a clearer and more concise paragraph that still contains that relevant information for readers. We refrain from adding too much more into the paragraph about how the data was compiled because there is already a lot of detail, and Price *et al.* 2012 already explores the justification for the methods used in compiling the dataset. Reviewer Grossnickle asked why 5% was the original cutoff for food presence or absence; it was because that was assumed to be a level that foods were regularly consumed by individuals even at lower proportions. To clarify, the Price *et al.* 2012 dataset does also have a number of species coded based on qualitative descriptions of diet, so even though many of the references do have proportional data within them, it is not completely based on proportional data. Additionally, proportions are not necessarily numerically equivalent between numerical data based on stomach contents, scat contents, or behavioral observations. We updated the text to communicate this combination of coding schemes. We also removed the section that talked about how proportional data would be better. We think this eliminates confusion about why we would think proportional data is better but do not use them, even though some of the original data is proportional.

We also added the sentence “The results of any diet study are determined in part by the choice of diet data and the way those data are standardized.” to acknowledge the uncertainty pointed out by reviewer Grossnickle, which we feel is applicable to all studies that incorporate ecological data from surveys of the literature, as the variation in methods, spatial scales, and temporal scales of the primary data collection yields a certain

level of uncertainty. We hope this addresses the concern that using a different dataset (with maybe a different definition for omnivory) could lead to a different result than our own.

Ln 117 added “by using functions” to the sentence about the caper package. This makes our point clear.

Ln 141-143 We clarified that we simplified the herbivorous dietary guilds into one because of consistency with the omnivore guilds and to compare our results with those of Price *et al.* 2012.

Ln 351-354 revised the last two sentences of the paragraph to clearly state that diet and evolutionary history are intertwined even for omnivorous mammals (which are over-dispersed on the tree). We feel that these sentences better communicate our point than our original statement “mammalian omnivory is tightly evolutionarily constrained”.

Ln 464-483 Given the feedback from the reviewer we edited the conclusion paragraph to have a stronger unified message. We removed sentences like: “Omnivorous mammals are diverse, have different evolutionary origins, and consume different foods.” We agreed that these sentences simplified our findings to a point where they lose their usefulness. We added sentences like “Our results show that even for omnivorous mammals, which are phylogenetically over-dispersed, diet, phylogeny, and body mass are connected.” We find these communicate a stronger message.

Ln We added D. Grossnickle and three anonymous reviewers to our acknowledgements.

For even more detail on how we edited our manuscript please see the word document with track changes. If there are questions or concerns, please let us know.

This manuscript has not been published elsewhere and is not under consideration by another journal. All authors have approved the manuscript and agree with its submission.

Thank you for your consideration,

Dana M. Reuter*, Samantha S.B. Hopkins, Samantha A. Price

Dana M. Reuter
Department of Earth Sciences
1272 University of Oregon, Eugene, OR 97403, USA
dreuter@uoregon.edu